# Intraseason variability of greenhouse gas emission factors from biomass burning in the Brazilian Cerrado

Roland Vernooij[1], Marcos Vinicius Giongo Alves[2], Marco Assis Borges[3], Máximo Menezes Costa[3], Ana Carolina Sena Barradas[3], Guido R. van der Werf[1]

[1]Department of Earth Sciences, Faculty of Science, Vrije Universiteit Amsterdam, Amsterdam, The Netherlands
[2]Center for Environmental Monitoring and Fire Management (CEMAF), Federal University of Tocantins, Gurupi, Brazil
[3]Chico Mendes institute for Conservation of Biodiversity (ICMBio), Rio da Conceição, Brazil

*Correspondence to*: Roland Vernooij (r.vernooij@vu.nl)

**Abstract.** Landscape fires, often referred to as biomass burning (BB), emit substantial amounts of (greenhouse) gases and aerosols into the atmosphere each year. Frequently burning savannas, mostly in Africa, Australia, and South America are responsible for over 60% of total BB carbon emissions. Compared to many other sources of emissions, fires have a strong seasonality. Previous research has identified the mitigation potential of prescribed fires in savanna ecosystems; by burning cured fuels early in the dry season when landscape conditions still provide moist buffers against fire spread, fires are in general smaller, patchier and less intense. While it is widely accepted that burned area (BA) and the total carbon consumed are lower when fires are ignited early in the dry season, little is known about the intraseason variability of emission factors (EF). This is important because potentially, higher EFs in the early dry season (EDS) could offset some of the carbon benefits of EDS burning. Also, a better understanding of EF intraseason variability may improve large-scale BB assessments, which to date rely on temporally-static EFs. We used a sampling system mounted on an unmanned aerial vehicle (UAV) to sample BB smoke in the Estação Ecológica Serra Geral do Tocantins in the Brazilian states of Tocantins and Bahia. The protected area contains all major Cerrado vegetation types found in Brazil, and EDS burning was implemented on a large scale since 2014. Over 800 smoke samples where collected and analyzed during the EDS of 2018 and late dry season (LDS) of 2017 and 2018. The samples were analyzed using cavity ring-down spectroscopy and the carbon balance method was used to estimate $CO_2$, CO, $CH_4$, and $N_2O$ EFs. Observed EF averages and standard deviations where 1651 ($\pm$ 50) g kg$^{-1}$ for $CO_2$, 57.9 ($\pm$ 28.2) g kg$^{-1}$ for CO, 0.97 ($\pm$ 0.82) g kg$^{-1}$ for $CH_4$ and 0.096 ($\pm$ 0.174) g kg$^{-1}$ for $N_2O$. Averaged over all measured fire prone cerrado types (i.e. excluding humid grasslands and gallery forest samples), the modified combustion efficiency (MCE) was slightly higher in the LDS (0.961 vs 0.956) and the CO and $CH_4$ were 10% and 2.3% in the LDS compared to the EDS. However, these differences were not statistically significant using a two-tailed t-tests with unequal variance at a 90% significance level. The seasonal effect was larger in more wood-dominated vegetation types. $N_2O$ EFs showed a more complex seasonal dependency, with opposite seasonal trends for savannas that were dominated by grasses versus those with abundant shrubs. We found that the $N_2O$ EF for the open cerrado was less than half the value in EF compilations for savannas. This may indicate a substantial overestimation of the contribution of fires in the $N_2O$ budget. Overall, our data implies that in this region, seasonal variability in greenhouse gas emission factors may offset only a small fraction of the carbon mitigation gains in fire abatement programs.

## 1. Introduction

Landscape fires emit large amounts of greenhouse gases and aerosols, which significantly impact atmospheric chemistry and biogeochemical cycles on local to global scales (Andreae and Merlet, 2001; Reid et al., 2005; van der Werf et al., 2017). The primary greenhouse gases emitted from biomass burning are carbon dioxide ($CO_2$), methane ($CH_4$) and nitrous oxide ($N_2O$). Over the period 1997–2016, average annual emissions from landscape fires were 7.3 Pg $CO_2$, 16 Tg $CH_4$, and 0.9 Tg $N_2O$ according to the Global Fire Emissions Database (GFED4s, Van der Werf et al., 2017). Tropical savannas accounted for the majority of these global landscape fire emissions with 4.9 Pg $CO_2$, 6 Tg $CH_4$, and 0.6 Tg $N_2O$. South American savannas on average accounted for about 10% of the global fire-related carbon emissions in savannas, corresponding to 6.5% of the total fire-related carbon emissions over this period. In general, biomass burning $CO_2$ emissions are compensated for by regrowth of vegetation after the fire (Beringer et al., 2007; Landry and Matthews, 2016). Therefore, fires only impact long-term atmospheric $CO_2$ concentrations when regrowth does not take place (e.g., following deforestation and tropical peatland fires) or if fire regimes become more severe and result in regional carbon sources. Although the IPCC recommends national accounting of indirect GHGs like CO and NMHC from biomass burning (Goodwin et al., 2019), carbon monoxide (CO) is generally not considered in discussions on emission abatement schemes (Australian Government - Department of the Environment and Energy, 2018; Cook et al., 2015; Lipsett-Moore et al., 2018). Like $CH_4$, CO reacts with atmospheric OH radicals and is eventually oxidized to $CO_2$ (Crutzen and Zimmermann, 1991; Daniel and Solomon, 1998). The depletion of OH radicals by enhanced CO concentrations leads to an increase in the atmospheric lifetime of $CH_4$ and the formation of ozone ($O_3$) (Crutzen and Zimmermann, 1991; Fry et al., 2012; Sudo and Akimoto, 2007). Albeit poorly understood, CO and $CH_4$ indirectly affect clouds by altering the abundance ofoxidants which convert $SO_2$ into sulfate (Penner et al., 2006). Therefore, CO can be viewed as an indirect greenhouse gas, more potent than $CO_2$ (Daniel and Solomon, 1998; Myhre et al., 2013).

Emission factors (EFs) are used to quantify the conversion of the total amount of carbon and other elements in the consumed fuel to emissions of various trace gases and aerosols. They are often reported in grams per kg of dry biomass consumed. Biomass burning EFs are derived from laboratory, ground-based, and aircraft measurements and have been reported for a large number of chemical species and vegetation types (Akagi et al., 2011; Andreae, 2019; Andreae and Merlet, 2001). The modified combustion efficiency (MCE), defined as the amount of carbon emitted as $CO_2$ divided by the amount of emitted carbon in $CO_2$ and CO combined, is often used as an indication of the relative contribution of flaming and smouldering combustion (Akagi et al., 2011). The MCE ranges from about 0.65 in smouldering peat fires to values close to one for highly efficiently oxidising grass fires. The negative correlation of MCE with EFs for incomplete combustion products such as $CH_4$, non-methane hydrocarbons (NMHC) and carbonaceous particulate matter (CPM) (Hoffa et al., 1999; Urbanski, 2013), makes it a useful metric for emission estimations.

A substantial amount of research has been conducted to understand what environmental factors affect the EFs for greenhouse gases (e.g., Chen et al., 2010b; Korontzi et al., 2003a; Urbanski, 2014). While the drivers of variability in CO and $CH_4$ EFs have received considerable attention, relatively little is known about the BB contribution to the $N_2O$ budget. $N_2O$ is formed through the oxidation of HCN and $NH_3$, in which the reaction of HCN through NCO is the dominant pathway. The $N_2O$ EF is

strongly dependent on the C:N ratio in the fuel (Lobert and Warnatz, 1993) as well as the temperature and partial pressure of oxygen during combustion (Kilpinen and Hupa, 1991; Winter et al., 1999a). $CH_4$ is formed during incomplete oxidation of biomass, with higher EF when fuels are relatively moist (Chen et al., 2010), or when fuels are densely packed (Bertschi et al., 2003; Urbanski, 2013). While some of the drivers of variability in these EFs are qualitatively known, large-scale studies have so far relied on biome-average estimates due to the lack of quantitative information, thus ignoring spatio-temporal variability

within biomes (Van Leeuwen and Van der Werf, 2011).

The Cerrado in South America consists of a mosaic of grasslands, shrublands, and forests. The biome covers roughly 24% of Brazil, as well as smaller parts of Paraguay and Bolivia (Klink and Machado, 2005). Vegetation dynamics and distribution in wild Cerrado areas are primarily determined by water availability, soil type and fire history (Pivello, 2011). The Cerrado can

be categorized based on the abundance of woody species, ranging from campo limpo (open grassland), campo sujo (grassland with sparse presence of shrubs), campo rupestre (rock field), parque cerrado (grass/shrub-dominated with scattered trees), cerrado típico (tree-dominated with scattered shrubs and a grass understory) to cerradão (forest savanna). Forested landscapes, such as gallery forests which tend to directly line the river and are found in riparian zones within the Cerrado, are particularly fire-sensitive (Ribeiro and Walter, 1998). Humid grasslands are also found here, consisting of gleysols that remain flooded in

the rain season which are typically covered with grass and sparse palm trees. Fires have a dominant role in limiting the proportion of trees in the Cerrado, and fire frequencies generally range from 3 to 8 years (Fidelis et al., 2018). Natural fires in the Cerrado are caused by lightning and mostly occur at the beginning and end of the dry season. Anthropogenic fires, lit for example for cattle ranging pasture improvement, typically occur around the middle of the dry season in July-August (Pivello, 2011; Ramos-Neto and Pivello, 2000; Schmidt et al., 2018). Fire intensity is a key landscaping factor that can also feedback

on longterm vegetation state; high-intensity fires limit tree cover and promote open grassland formation, which in turn promotes higher fire frequency (Miranda et al., 2009; Oliveras et al., 2013; Staver et al., 2011a).

Under seasonally dry conditions, Cerrado grass species dry out and senesce, leading to standing dead fuel accumulation (Fidelis and Fernanda, 2013). Local practices have relied on prescribed burning in the past, and scientific research showed that fire has

a key role in maintaining the Cerrado's high biodiversity. However, until the first integrated fire management approach for some protected areas was launched in 2014, a 'zero-fire' policy had been maintained in the Brazilian Cerrado for decades (Durigan et al., 2016; Schmidt et al., 2018). While fire suppression strategies can be effective as a tool to enhance carbon sequestration and total carbon stocks (Murphy et al., 2010; Staver et al., 2011a), keeping fire out of the Cerrado altogether potentially leads to a sharp decline in biodiversity through the loss of light-demanding savanna species (Abreu et al., 2017;

Durigan et al., 2016). In larger continuous landscapes, fire suppression strategies have led to a shift towards more high-intensity late dry season (LDS) fires which are more difficult to suppress. Frequent, high-intensity fires can cause long-term losses of soil nitrogen and phosphorous (Kauffman et al., 1994), which in turn decreases the total amount of carbon that is sequestered by net primary productivity. This may in time alter the carbon sink capacity of frequently burning savanna grasslands (Pellegrini et al., 2018). To combat the rise of intense LDS fires, it is important to look for alternative fire management strategies. Somewhat ironically, fire exclusion experiments have thus shown that well-managed fire intervals and intensities are vital for sustaining biodiversity in fire-prone savanna systems (Abreu et al., 2017; Durigan et al., 2016; Scott et al., 2012).

Given that fire exclusion and thus a fire-free Cerrado is hardly possible nor beneficial, previous studies have suggested the potential for prescribed burning in the early dry season (EDS) as an alternative to devastating LDS fires (Fidelis et al., 2018). Cerrado fuels dry at different rates under different landscape conditions (e.g. slopes vs valley bottoms) as the EDS progresses, resulting generally in smaller, more patchy, less intense fires (Rissi et al., 2017). Networks of strategically placed EDS fires can be used to reinforce natural (e.g. riparian corridors) and built (e.g. road) barriers, thereby reducing the risk of more intense, extensive LDS wildfires. For this reason, prescribed EDS burning is suggested as a climate mitigation strategy in the savanna (Anderson et al., 2015; Lipsett-Moore et al., 2018; Penman et al., 2011; Russell-Smith et al., 2013).

Africa and South America combined collectively account for about 65–77% of the fire-prone savannas, and carbon-schemes may provide incentivised alternatives for delivering less destructive savanna fire regimes as shown in Australia (Lipsett-Moore et al., 2018; Maraseni et al., 2016; Russell-Smith et al., 2013). Wildfire emissions are the product of fire extent, fuel load, combustion completeness, and EFs for the emitted species (Seiler and Crutzen, 1980). EDS fires have been demonstrated to be smaller and more patchy (Oliveira et al., 2015; Price et al., 2012), and result in lower combustion completeness (Yates et al., 2015). Total fuel consumption is therefore lower. However, through more incomplete combustion under more humid fuel conditions, higher $CH_4$ EFs offset some of the emission gains from reduced fuel consumption (Hoffa et al., 1999; Ito and Penner, 2004; Korontzi, 2005; Van Leeuwen and Van der Werf, 2011; Yokelson et al., 2011). Understanding and quantifying the intraseason variability in EFs is therefore essential to assess the implications of natural- and human-induced fire regime shifts.

In this study we have used a novel unmanned aerial vehicle (UAV)-based approach to sample fires during three field campaigns, covering different parts of the dry season and various fire-prone Cerrado vegetation types. Our main objective was to assess the spatio-temporal variability in EFs for the main greenhouse gases associated with BB. With this knowledge we are in a better position to understand the carbon mitigation potential of savanna fire management and these findings may improve the representation of EFs in large-scale fire databases.

## 2. Methodology

### 2.1 Study area

The Estação Ecológica Serra Geral do Tocantins (hereafter referred to as EESGT) is a protected area located in the Brazilian states of Tocantins and Bahia (Fig. 1a). Covering ~700,000 hectares, it is one of Brazil's largest ecological stations; a type of strict protected area established to preserve representative samples of the different biomes in Brazil. EESGT used to be one of the most frequently burning protected areas in the Cerrado. On average, about 30% of the protected area burned each year (Fidelis et al., 2018). Since 2014, prescribed EDS burning has been used within EESGT as a tool to reduce the negative impacts from uncontrolled, high-intensity LDS fires. The strategy focusses on creating a mosaic of smaller areas with different fire histories hence varied fuel loads (Barradas et al., 2020). Since 2014, the strategy has resulted in an increase in the number of fires but a decrease in average fire size and total burned area (Fidelis et al., 2018; Schmidt et al., 2018). This is in contrast to other protected areas in the Cerrado, which have implemented limited prescribed burning and experienced an increase in burned area over the same period. For example, 78% of nearby Chapada dos Veadeiros, and 85% of Reserva Natural Serra do Tombador were burnt in 2017, inciting calls for wider implementation of EDS fire management (Fidelis et al., 2018).

EESGT has a semi-arid to tropical climate with two well-defined seasons. Hot and dry in May to September, and wet and cooler in the rainy season between October and April (Fig. 2a). With an average annual rainfall of around 1400-1500 mm, EESGT is somewhat wetter than the Cerrado average of 1300 mm (Seplan, 2003). The area is dominated by nutrient-poor, deep arenaceous quartz soils and has a high floristic biodiversity. All the major Cerrado vegetation types are represented in the ecological station, but the area is dominated by open grasslands (capo limpo and campo sujo) and open savanna vegetation (cerrado ralo and cerrado típico/sensu stricto) (Fig. 1b).

The fire season in EESGT roughly lasts from May until October and peaks around September (Fig. 2a). In the EDS, managers apply fires during a 'safe-burning window' which depends on the vegetation type, fuel conditions, and weather. Typically, EDS prescribed fires are applied in the afternoon, and extinguish after sunset as temperature and wind speed drop and relative humidity increases. Managers actively suppress intense LDS fires to protect vulnerable vegetation and surrounding communities (Barradas et al., 2018).

### 2.2 Measurement campaigns

We carried out one EDS- and two LDS measurement campaigns. During the 2017 LDS (23 Sept - 11 Oct), fires were ignited between 9:30 and 18:00 and air temperature ranged from 25-38°C. No relative humidity measurements were taken during this campaign. In the EDS of 2018 (16-30 June), fires were ignited between 12:00 and 18:00. The air temperature during this period, as measured prior to the fire at an elevation of 15m, was between 31 and 36 °C with an average relative humidity of 18%. During the 2018 LDS (23 Sept - 11 Oct) samples were collected from prescribed and two non-prescribed fires, in which

we sampled smoke from gallery forest and humid grasslands fires during a LDS fire repelling effort. Although the LDS campaign in 2018 ocurred after the first rainfall of the season, which came early in 2018, fire intensities appeared to be much higher than during the EDS campaign (Fig. 3). The temperature ranged from 37-42 °C with an average relative humidity of 13%. More information about the number of measurements taken, and vegetation type coverage during each campaign, is listed in Table 1.

## 2.3 Sampling strategy

We filled single-polypropylene fitted Tedlar bags (SKC, type 232-01) with fresh smoke using a UAV-based (DJI, Matrice 100) sampling system. Most of the samples were taken 15-20 meters above the fire, with the height increasing with intensity of the fire. Our sampling system consisted of a container mounted on top of the UAV which contained four Tedlar sample bags. We filled 1-liter bags with ±0.7 liters of smoke, which took 35 seconds for each bag creating a 35s-averaged mixture of trace gases in the bag. The sample inlet was located on the top of the UAV and fitted with a 60-µm sintered porous metal filter. During the sampling period, the system logged time, GPS coordinates, pressure, temperature and relative humidity on the UAV.

Most sampled fires were ignited by the EESGT rangers using a drip-torch to start a fire line of at least 30 meters. We started sampling when the fire-front had advanced 10-20 meters. We positioned the UAV to capture a mixture of the fast-ascending flaming combustion products and the smouldering products that were generated upwind from the flaming fire-front. While the majority of the fires sampled were prescribed burns, we also sampled several non-prescribed LDS fires. These fires were most likely escaped pasture fires or poaching fires, given that lightning did not occur during our LDS campaign. We sampled both EDS and LDS fires over various vegetation types with the time since the last fire ranging between 2 and 5 years (Table 1).

## 2.4 Smoke analysis

We used cavity ring-down spectroscopy to analyze $CO_2$, CO, $CH_4$, and $N_2O$ concentrations from the sample bags. After sampling, the Tedlar bags were kept in a dark environment and analyzed within 12 hours. This was done in order to minimize the oxidation of CO by OH radicals inside the bags. According to Meyer et al. (2012) and our own tests, $CO_2$, CO, $N_2O$, and $CH_4$ concentrations are stable in the Tedlar bags for extended periods under these conditions. The samples were measured for 20 seconds at a flowrate of 1.3 L min$^{-1}$ using a $CO_2$ and $CH_4$ analyzer (Los Gatos Research, Microportable) followed by 20 seconds at a flowrate of 0.25 L min$^{-1}$ using a CO and $N_2O$ analyzer (Aeris Technologies, Pico series), see Table 2. Measurement of the trace gas concentration in the bags was based on the 10s average concentration following a 10s initial flushing period. Before each fire, we filled four "background" samples at 15 meter altitude. The average concentration of these background samples was subtracted from those in the plumes to get the excess mixing ratio (EMR) in the sample bags. Variability between the background samples during a single day was smaller than 5%. Both analyzers were calibrated before and after each campaign using certified standard calibration gas (Table 2). No significant calibration drift was observed during the campaigns.

## 2.5 EF calculation

We converted the EMR (sample minus background concentration) in the bags to EFs for $CO_2$, CO, and $CH_4$ in grams of emitted species per kilogram of dry matter burned, following the carbon mass balance method (Urbanski, 2013; Yokelson et al., 1999):

$$EF_i = F_c \times \frac{MW_i}{AM_c} \times \frac{C_i}{C_{total}} \times 1000 \; g \; kg^{-1} \tag{1}$$

Where $EF_i$ is the emission factor of species $i$ and $Fc$ is the carbon content of the fuel by weight fraction. In this study, we used 48% for grassland/savanna and humid grasslands and 50% for gallery forest, based on carbon content measurements from different cerrado vegetation types by Susott et al. (1996). $MW_i$ is the molecular mass of species $i$ divided by the atomic mass of carbon ($AM_c$). $C_i$ is the number of moles of carbon emitted in species $i$, $C_{total}$ is the total number of moles of emitted carbon. Because we did not measure non-methane hydrocarbons (NMHC) and carbonaceous particulates (CPM), these

fractions were estimated based on ratios from savanna burning literature. The total mass of emitted CPM was estimated to be 7% of the emitted mass of CO (Andreae, 2019), with carbon accounting for 70% of the CPM-mass (Reid et al., 2005). The total amount of carbon in NMHC was estimated to be 3.5 times the $ER_{(CH4/CO2)}$ based on common ratios for savanna fires (R.J. Yokelson, personal correspondence). We did not consider residual ash in our calculations which can represent significant amounts of carbon (Jones et al., 2019). Although this is common practice in EF calculations, leaving out ash may lead

to overestimation of carbon emissions (Surawski et al., 2016). To calculate the EFs for $N_2O$, we used Eq. (2) described by Andreae and Merlet (2001). This method uses the emission ratio ($ER_{(i/y)}$) of the species $i$ to a relatively inert, co-emitted carbonaceous species $y$.

$$EF_i = ER_{(i/y)} \times \frac{MW_i}{MW_y} \times EF_y \tag{2}$$

We used $CO_2$ as the co-emitted reference gas following earlier work (Hao et al., 1991; Hurst et al., 1994a; Surawski et al.,

2015). Although CO is also sometimes used for this purpose due to its low background variability (Meyer et al., 2012), based on previous continuous emission measurements, we found $N_2O$ to be more closely correlated with $CO_2$. Calculation using CO as a co-emitted reference gas for the $N_2O$ EF on average would lead to $N_2O$ EFs that are 2.4% higher. EFs were calculated for each bag separately, and we partitioned the bags into different seasonal, vegetation type, and fire history classes (Sect. 2.6). To get the weighted average EF for these classes, we calculated EFs over the cumulative EMR of the respective trace gas

species from all samples in the class. Samples with low overall trace gas concentrations thus have low impact on the weighted average EF.

To assess the intraseasonal effect of the emissions on radiative forcing (RF), we calculated the $CO_2$-eq EF based on the EFs weighted by the 100-year Global Warming Potentials ($GWP_{100}$) if the emitted species. $CH_4$ and $N_2O$ have a $GWP_{100}$ of 34 and 298, respectively, when climate-carbon feedback mechanisms are included (Myhre et al., 2013). CO is usually not included

but through its removeal if hydroxide it leads to a longer lifetime of $CH_4$, is a precursor for $O_3$, and eventually oxidizes to $CO_2$

resulting in 1.57g $CO_2$ for each gram of oxidized CO (Goodwin et al., 2019). Therefore we have also taken CO into account. Estimates of the indirect CO $GWP_{100}$ vary from 1.8 (Fry et al., 2012) to 5.4 when taking into account primary and secondary aerosol effects on clouds (Shindell et al., 2009). We used a $GWP_{100}$ for CO of 2.2 which is on the conservative side of these estimates and does not include the effect of the oxidized $CO_2$, since this is assumed to be compensated for by regrowth.

**2.6 Spatial analysis and upscaling**

All samples were geolocated using the coordinates of the UAV. This location was used to tag the samples with vegetation type and the number of years since the previous fire derived from satellite data. Most of the plumes were sampled close to the fire, but we manually checked this information with satellite BA data to avoid mismatches due to plume advection. To calculate the fire history of the burned vegetation we used 30m Landsat thematic mapper (TM), enhanced thematic mapper (ETM) and

operational land imager-based BA data from the Instituto Nacional de Pesquisas Especiais (INPE) (Melchiori et al., 2015). The dataset uses consecutive Landsat scenes to detect changes in Normalized Difference Vegetation Index (dNDVI) and Normalized Burn Ratio (dNBR) for BA classification. The BA classification is manually validated in the field and thresholds in the algorithm were optimized for EESGT as described by Barradas et al. (2018). The number of years since the last fire was determined based on the location of the sample and the Landsat 30m burn-scars of the last years.

For the vegetation classification described in Table 1, we used maps created by the University of Brasilia (Fig 1b), which were derived from 5m RapidEye multispectral imagery (Orozco-Filho, 2017). The classification is based on spectral characterization of the different vegetation types and distinguishes the following Cerrado classes sampled by this study: campo limpo/sujo (open grassland; 0-5% tree cover), cerrado *sensu stricto* ralo (open cerrado; 5-20%), cerrado *sensu stricto* típico (tipical cerrado; 20-50%), cerrado denso (dense cerrado; >50%), gallery forest (continuous canopy) and riparian zones (sparse palm

trees in wetlands). The classification matched well with our field observations during the campaigns but we did not validate the map formally. It should be noted that the fractional tree cover (FTC) classification in the RapidEye map generally leads to higher FTC values compared to the Moderate Resolution Imaging Spectroradiometer (MODIS) based vegetation continuous fields dataset (MCD44Bv6, Townshend et al., 2011), or the Landsat-based rescaling of the MCD44Bv6 dataset (Sexton et al., 2013). Hence, care should be taken with spatial extrapolation of these vegetation classes using different FTC products.

The weighted average ($\overline{EF}$) for combined Cerrado vegetation types in the EESGT was calculated through Eq. (3) in which $n$ is the number of vegetation types, $BA_i$ is the burned area based on the aforementioned EESGT-optimized INPE BA (Barradas et al., 2018), for the vegetation class ($i$) over the years 2013 to 2018 and $BA_{tot}$ is the total burned area in over the same period (Fig. 4).

$$\overline{EF} = \sum_{i=0}^{n} EF_i \times \frac{BA_i}{BA_{tot}} \tag{3}$$

Since we lack data on the fuel load and combustion completeness, we weighed the EFs by the percentage of BA in the different classes (Fig. 4). Further, given that we do not have measurements of dense cerrado EFs, the dense cerrado BA was accounted for as typical cerrado. As the BA composition of EDS fires primarily depends on management considerations, both seasons were weighed by the total averaged BA composition.

## 3. Results

The weighted average EFs for the different vegetation types, as well as the $\overline{\overline{EF}}$ for combined Cerrado vegetation, are listed in Table 3. Since the introduction of prescribed LDS burning in EESGT in 2014, the proportion of BA before the first of July has gradually increased (Fig. 2b). This has been the case for all dominant fire-prone vegetation types found in EESGT. As the samples were unevenly distributed over the different vegetation types (Table 1), we had to account for the sample bias in vegetation type to compare EDS and LDS EFs, our main objective. To obtain a seasonal weighted-average emission factor ($\overline{EF}$) for Cerrado vegetation, we therefore weighted the different cerrado vegetation class EFs by their contribution to the fires in EESGT. Over the 2013-2018 timeframe, the distribution of BA over the different fire-prone ecosystems (vegetation types) most common in the EESGT was approximately 23% in open grassland, 42% in open cerrado, 28% in typical cerrado, and 7% in dense cerrado.

### 3.1 Intraseasonal variability

Although the variability within individual fires (we collected several samples from each fire), vegetation types and campaigns was high, the difference between the season-averaged CO and $CH_4$ EFs was limited (Fig. 5). The MCE increased slightly from a weighted average of 0.957 in the EDS to 0.963 in the LDS. When considering individual vegetation types, more efficient combustion in the LDS campaigns is apparent. For example, the difference between the LDS and EDS when averaged over all vegetation types (-15% for CO and -13% for $CH_4$) is more pronounced when focusing on more shrub-dominated areas (open cerrado). For example, CO and $CH_4$ EFs were 18 and 21% lower in the LDS for typical cerrado vegetation (Table 3). As a result of the large spread in EFs and a limited number of samples in some vegetation types, only the slight differences in open grasslands and the 14% and 34% increases in $N_2O$ EF for open cerrado and typical cerrado, respectively, were statistically significant using a two-tailed t-tests with unequal variance at a 90% significance level.

Campaign-averaged $N_2O$ EFs were 0.105 g $kg^{-1}$ in the EDS and 0.123 g $kg^{-1}$ in the LDS. However, internal variability within the campaigns was high with standard deviations of 0.183 g $kg^{-1}$ in the EDS and 0.263 g $kg^{-1}$ in the LDS. In Table 3, $N_2O$ EFs are reported for samples with enhanced carbon concentrations of over 15 moles (as explained in Sect. 4.4), in order to minimize propagation of measurement error in the standart error of the mean as explained in Sect. 4.4. Though not significantly altering the weighted average, this improved the significance of the found relationships. In Figs. 5-7 the green diamond represents the arithmetic mean and the red cross represents the EMR-weighted mean. Measurements more than 1.5 times the interquartile

range (IQR) above the upper or below the lower quartile are presented as outliers (open circles). Whiskers represent the outermost values within 1.5 times the IQR of the respective quartiles. The third boxplot represents the spread in EFs from different studies on BB EF in savannas, and the value that is currently used in large scale emission assessments. If we investigate the $N_2O$ EF intraseasonal variability within the vegetation type classes, we find opposite trends (Table 3). In the open grasslands (campo limpo/campo sujo), the weighted average $N_2O$ EF in the EDS was more than double the $N_2O$ EF in the LDS. In the open cerrado (cerrado ralo) and typical cerrado (cerrado típico), however, the weighted-average $N_2O$ EFs were 14% and 34% higher in the LDS.

### 3.2 EF variability in vegetation type and fire history

We found no significant differences in the MCE, CO EF and $CH_4$ EF between the EMR-averaged values of the different Cerrado vegetation types, despite substantial differences in tree cover density (Fig 6). The samples we took over gallery forest contained much higher EFs for CO and $CH_4$, indicating more smouldering combustion. The $N_2O$ EF was found to be positively correlated with tree cover and was a factor 5 times higher in the gallery forest compared to savanna vegetation.

Fire history had some effect on the burning efficiency. We found a decrease in the CO EF and $CH_4$ EF (and thus increase in MCE) with increasing time between fires ranging from 2 to 4 years in samples from the open grasslands (Fig. 7). Although the measurements in typical cerrado did not cover the entire fire-frequency span, the available data suggested no significant relation between EFs and the years since the last fire in both open cerrado- and typical cerrado vegetation (not shown).

### 3.3 GWP variability between EDS and LDS fire

Fig. 8 shows the cumulative $CO_2$–equivalent (eq) of the respective gases, based on a 100-year time span. Overall, $CO_2$-eq emissions per kg of dry fuel in the Cerrado were 8.2% lower in the LDS compared to the EDS. The difference between EDS and LDS $CO_2$-eq can largely be contributed to somewhat more efficient combustion in the LDS which is partially compensated for by a higher $N_2O$ EF. The black error bar represents the propagation to the net $CO_2$-eq emissions of the combined standard error of the mean of all species. 12% to 50% of this error comes from the propagation of the uncertainty in $N_2O$ EFs. Even without taking aerosol effects into account, the indirect radiative forcing due to CO made up a significant portion (45-65%) of total $CO_2$-eq emissions.

## 4. Discussion

### 4.1 Difference in EFs between EDS vs LDS fires

Korontzi et al. (2003) found that the seasonal curing cycle affected MCE in prescribed burn plots in southern African savannas. This intraseasonal shift would limit or even cancel climate benefits of EDS prescribed burning. They found that for 'Dambo' grasslands, EFs for reduced species were strongly correlated with the percentage of green grass in the fuel. This percentage

decreases as grasses cure over the course of the dry season. A similar trend was found by Yokelson et al. (2011) when comparing EF measurements for EDS fires in Mexico to LDS African savanna measurements. Direct measurements taken during the West Arnhem Land Fire Abatement Project (WALFA) in northern Australia, however, showed no significant seasonal fluctuation in both $CH_4$ and $N_2O$ EFs (Hurst et al., 1994b; Meyer et al., 2012). Measurements taken in Zambian miombo woodlands did not show significant seasonal MCE fluctuation either (Hoffa et al., 1999).

In this study we measured EFs during lower-intensity fires in the EDS as well as higher intensity LDS fires, all in the same region. Although we also found some intraseasonal difference, the decrease of EFs for CO (-15%) and $CH_4$ (-13%) was small compared to the -68% (for CO) and -81% (for $CH_4$) found for African grassland fires (Korontzi et al., 2003a). In addition, intraseasonal variability was smaller compared to the variability within EDS or LDS campaigns, and the difference was not statistically significant (p<0.1). The average $N_2O$ EF over the combined Cerrado samples showed a slight increase over the season, though stronger and opposing seasonal trends were found in the individual vegetation classes. Meyer et al. (2012) also found opposing seasonal $N_2O$ EF trends for different vegetation types. While the formation process of $N_2O$ is often linked to combustion characteristics (Kilpinen and Hupa, 1991; Meyer et al., 2012; Winter et al., 1999a), we did not find a significant correlation of the $N_2O$ EF with MCE. Overall, MCE values were higher than the average MCE values derived from $CO_2$ and CO EFs for savanna and grassland fires in Andreae (2019), but within the range of previous measurements from Cerrado vegetation (Ferek et al., 1998; Ward et al., 1991). Over all Cerrado vegetation types combined, the weighted average $CH_4$ EF slightly declined over the dry season.

We conducted the EDS experiments in June when the majority of the prescribed burning takes place (Fig. 2a). Although the LDS measurements in 2018 were taken after the first rains, conditions were still hotter and dryer than during the EDS, and the combustion completeness appeared to be higher (Fig. 3). No fuel moisture measurements were done during the 2018 campaigns but co-located measurements from 2017 showed limited drying occurring from June to September, with respective average fuel moisture contents (FMC) declining from 63.8% to 55.4% for live grass and 11.7% to 7.2% for dead grass (Santos et al., in press). Larger differences may be expected earlier in the EDS period March-May (N'Dri et al., 2018), when the FMC and live-to-dead fuel ratio is even higher (Santos et al., in press). During these months, when humidity is still very high, prescribed burning efforts focus on the protection of vulnerable ecosystems such as peatlands and gallery forests, as well as areas around homes and farmlands, but total BA is limited. Additional measurements in the very start of the dry season (March-May) should confirm whether EFs increase for these fires. Rissi et al. (2017) measured fuel characteristics, rate of spread, flame height, fire intensity (kW m$^{-1}$) and combustion completeness in campo sujo vegetation (<20% tree-cover) for prescribed burns in May, July and October. Although the spread in fire intensity between fires was higher in the late season, they found no significant differences in these charactetristics between the July and August treatments. Fire intensity was best explained by fuel build-up (Rissi et al., 2017); this is consistent with the MCE increase we found between 2-4 years of fuel build-up in open grassland vegetation (Fig. 7). The finding that the number of years since the last burn did not significantly affect the combustion

efficiency after 4 years is consistent with the results from Govender et al. (2006). However, we only found this correlation in open grassland with annual grasses leading to accumulation of easily combustible dead biomass.

## 4.2 Variability in CO and CH$_4$ EFs

According to our results, there was no significant difference in CO and CH$_4$ EFs between the dominant savanna vegetation types in EESGT: campo limpo/sujo, cerrado ralo, and cerrado típico. Overall, the weighted average CO and CH$_4$ EFs for these combined savanna fuel types were lower than most of the existing literature on savanna fires (Akagi et al., 2011; Andreae, 2019) (Fig. 5). The discrepancy with literature is particularly strong for CH$_4$ as shown in Fig. 9 where the individual CH$_4$ EF measurements are plotted as a function of MCE measured for the Cerrado vegetation types. Results from other studies, plotted as the study-averages, are shown based on the individual papers included in Andreae (2019). The averaged EFs were rather similar between EDS and LDS campaigns, but within each campaign, the EFs varied substantially. The shift in the LDS towards a steeper slope of the CH$_4$ EF/MCE linear regression in Fig. 9 may be an indication of a shift toward more combustion of woody fuels (Van Leeuwen and Van der Werf, 2011). Although the lower CH$_4$ EF found in this study can partially be explained by on average higher MCE values in our plots, the CH$_4$ EFs were much lower than average CH$_4$ EFs from savanna literature studies with the same MCE. Within the total range of variability, the slopes of the linear regression we found for both EDS and LDS campaigns were significantly less steep compared to the regression slope based on previous measurements of savanna vegetation CH$_4$ EFs. This is to some degree surprising given that the relation between MCE and CH$_4$ is thought to be well understood. In part, the lower slope comes from a larger number of earlier observations in the 0.90-0.95 MCE range; in the higher MCE ranges our results deviate less from earlier work. This may indicate that there is more variability in fire processes between different savanna types than previously reported. Also compared to earlier measurements from Cerrado vegetation the CH$_4$ EFs were low; Ferek et al. (1998) found an average CH$_4$ EF of 3.7 g kg$^{-1}$ and Ward et al. (1992) found CH$_4$ EFs ranging from 1-1.6 g kg$^{-1}$. This indicates that more research is needed over ideally a larger range of Cerrados and regions to understand what drives this variability.

The difference between EDS and LDS weighted average CH$_4$ EFs is partly the result of a larger spread and high-concentration of residual smouldering combustion (RSC) samples in the LDS (Fig. 5). Although the CH$_4$ EF was lower in the LDS (-13%, Table 3), the overall spread of CH$_4$ EFs in the LDS fires was higher than during EDS fires. Moreover, during the EDS, high CH$_4$ EFs are mostly found in samples with low overall trace gas EMRs (Fig. 10), meaning their impact on the EMR-weighted average was small. An explanation for the increased spread of CH$_4$ EFs in the LDS when the relative humidity was lower may be the effects of more complete combustion of grasses and fine fuels on one hand, and an increased share of RSC-prone woody fuels in the fuel mixture leading to a higher CH$_4$ EF on the other hand (Bertschi et al., 2003, Hoffa et al., 1999). These fuels typically contain more moisture in the EDS and are densely packed; therefore, they are more likely to burn in the LDS when humidity is low (Akagi et al., 2011; Eck et al., 2013; van Leeuwen et al., 2014). This is also observed in Australian savannas,

where combustion completeness of woody debris was found to be twice as high in the LDS compared to EDS fires (Yates et al., 2015).

Savanna areas with higher tree cover had slightly higher EFs for $N_2O$. Furthermore, there was an opposite seasonal trend in $N_2O$ EFs from grass-dominated campo limp/campo sujo (-55% from EDS to LDS) and shrub-dominated cerrado ralo (+14%) and cerrado tipico (+34%). Winter et al. (1999b) found $N_2O$ EFs to be closely correlated with the nitrogen content of the fuel. Susott et al. (1996) and Ward et al. (1992) measured the dry-weight carbon and nitrogen contents of various fractions of savanna fuels. For the Cerrado, they analysed dead- and living grass, dicots, litter, leaves and various woody debris fractions for the most fire-prone Cerrado classes studied in this paper. While they found that carbon-content in living grasses was only slightly higher compared to dead grasses, nitrogen content in living grass was on average more than double the content of dead grass. They also found that nitrogen contents of leaf, litter and dicot fractions increased in more woody vegetation types. The nitrogen content of coarse woody debris tends to decrease with the size of the debris. The opposite seasonal trends in $N_2O$ EFs may therefore be related to a seasonal shift in vegetation types that burn. Many shrubs and trees in EESGT are deciduous or semi-deciduous and drop all or part of their leaves throughout the dry season. This creates a fire-prone, nitrogen-rich litter layer that burns mostly in the LDS fires. In the open grasslands however, where leaf litter is not as significant to the fuel mixture, the ratio of dead versus living grasses increases which could reduce the nitrogen content of the fuel (Yokelson et al., 2011). The decline found in $N_2O$ EF from open grasslands that have not burned for some years (Fig. 7) may thus be related to the increased dead to live grass ratio of the fuel mixture as found by Santos et al. (in press). Whether this is indeed the explanation for the opposite seasonal trends in $N_2O$ emission factors requires future campaigns which include measurements of fuel load, combustion completeness, and nitrogen content over the whole season.

During the LDS, fires can escape into the peatlands and gallery forests lining the rivers. Many EF measurements in the savanna biome are conducted in research plots that are representative of the typical savanna vegetation. These plots, therefore, do not include EFs of these fine-scale landscape features. For this reason, we assessed them separately, and have not included them in the Cerrado weighted averages. Fires will only occur in these vegetation types in the LDS, when fuels are relatively dry and the groundwater table low. Late wet season management fires in these vulnerable vegetation types are used to reduce moribund fuels. Because we only took a few samples from gallery forest (26 samples) and humid grassland (15 samples), more research is needed in these vegetation types. Based on our measurements in the EESGT and relatively high N:C ratio of these ecosystems as described in literature, the $N_2O$ EF of 0.2 g kg$^{-1}$ currently applied in emission databases both for gallery forest ("tropical forest") and humid grasslands ("savanna") is likely a significant underestimate. The $CH_4$ EFs for humid grasslands tropical forests (Akagi et al., 2011) respectively.

### 4.3 Uncertainties

The main uncertainties associated with calculating fire-averaged EFs from field measurements include representativeness of the measurements taken related to the sampling strategy, measurement uncertainties, and assumptions based on other literature to represent factors not measured but required to compute EFs.

### 4.3.1 Sampling strategy

Given the unpredictable nature of fires and difficulties to move around during a spreading fire in a protected area without many roads, we tackled each fire differently. We could not standardize the strategy with regards to sampling head-, back-, and sideway propagating fires. Especially in the LDS fires, it was difficult to take many samples over the fast-moving fire front. Therefore, sideway propagating fires may be overrepresented in the dataset. According to Surawski et al. (2015) based on wind tunnel experiments, and Wooster et al. (2011) based on experimental field burns, fire spread mode affects EFs with, in general, slightly lower MCE occurring in headfires. Compared to earlier studies, we have taken a much larger number of samples thus lowering biases. To better calculate the representative mean requires better-contained fires that are easier to access and continuous sampling at various locations.

During the LDS, fires were predominantly sampled from 11:00 until 16:00 when temperatures are highest. However, these LDS fires generally last for multiple days, and measurements taken during the night and early morning are under-represented in the dataset. Diurnal fluctuations in temperature, wind, and humidity may cause these fires to behave more similarly to EDS fires during these times. Even though the amount of carbon consumed during those times is presumably lower than during the day, future efforts could shed light on the diurnal cycle of EFs.

An additional source of uncertainty stems from a potential bias related to sampling of RSC conditions. If the sampling period overlapped with the fire duration including the RSC, as was often the case for grasslands, derived EMR values are likely to have been representative. However, as RSC may persist also after we stopped sampling, especially in more woody fuels, EFs of predominantly RSC products such as CO and $CH_4$ may be underestimated using our sampling strategy. In Fig. 4 the difference between the arithmetic mean (green triangle) and the weighted mean (red square) represents the effect of weighing the bags by excess mixing ratio. In most cases, the difference is small, suggesting that the total contribution of RSC is limited. This is consistent with Ward et al. (1992), who measured BB emissions in Cerrado vegetation. They found that over 97% of the total carbon released was emitted during the flaming phase. The relatively low significance of RSC in grass-dominated savannas was also found for experiments in the Kruger national park, South Africa (Cofer et al., 1996; Wooster et al., 2011).

While the role of RSC in these grass-dominated ecosystems is thus thought to be small, the significance of RSC in areas with more woody fuel may be higher (Bertschi et al., 2003; Christian et al., 2007; Hao et al., 1991). With prescribed fire-

management, dead organic matter and woody carbon stocks may increase over time (Oliveras et al., 2013; Pivello, 2011; Veenendaal et al., 2018). For long-term emission abatement potential, it is therefore important to understand how these changes in fuel composition affect EFs.

### 4.3.2. EF calculations and assumptions

Ideally, EF calculations are based on measurements of all emitted carbon-containing species. This allows for the direct conversion of emission ratios to EFs per unit of burned fuel. We did not measure non-methane hydrocarbons (NMHC) and carbonaceous particulates (CPM). When combined, these can account for a significant portion of the total carbon emitted. To account for this, we have made assumptions for the $C_{NMHC}/C_{CH4}$ ratio (3.5, R.J. Yokelson, personal correspondence), the $PM_{2.5}/CO$ mass ratio (0.07) and the carbon fraction of $PM_{2.5}$ (0.70), based on Andreae (2019) and Reid et al. (2005). This adds

an additional 0.4-2.7% C from NMHC and 0.5-1.9% C from CPM to the total carbon balance. Most studies only include carbonaceous trace gases in the total carbon. However, leaving out part of the carbonaceous emissions artificially increases the EFs of the measured species. This inflation is proportional to the carbon that is not accounted for and will likely be in the 1-5% range (Akagi et al., 2011; Yokelson et al., 2013). EFs for both NMHC and PM are negatively correlated with combustion efficiency (Hoffa et al., 1999; Yokelson et al., 2013). Therefore, the overestimation of EF would be slightly larger in the EDS

compared to the LDS. As the $N_2O$ EF is coupled to a carbonaceous co-emitted species, in our case $CO_2$, this inflation will also affect the $N_2O$ EF.

Another source of uncertainty is the carbon content of the fuel. EFs scale linearly with this fraction and we used 48% for typical cerrado vegetation and humid grasslands and 50% for gallery forests (Susott et al., 1996). The carbon content in

humid grasslands is based on the assumption no peat, which has a higher carbon content of ~56% (Susott et al., 1996), was combusted in the fire. Had we made other assumptions, for example 45% (Andreae, 2019; Andreae and Merlet, 2001) or 50% (Akagi et al., 2011; Urbanski, 2014), our EF estimates would have been 6% lower to 4% higher in typical cerrado types and humid grasslands and 10% lower or equal in gallery forest. This scaling does not affect the spatial and temporal patterns we found.

### 4.3.3. $CO_2$-eq calculations and assumptions

Finding a useful metric to assess the direct and indirect impact on RF and climate is challenging, as mechanisms and time frames of the impact often differ between studies (Fuglestvedt et al., 2010). Atmospheric impact may also depend on geographic location, injection height or atmospheric conditions (Daniel and Solomon, 1998; Fry et al., 2012). There is substantial uncertainty in GWP (e.g. ± 40% for $CH_4$ $GWP_{100}$), dominated by uncertainty in the actual GWP for $CO_2$ (i.e. the

denominator of the GWP ratio) and inclusion of indirect effects (Myhre et al., 2013). We used an indirect $GWP_{100}$ for CO of 2.2; i.e. taking into account the $CH_4$ and $O_3$ effects but not considering primary and secondary aerosol effects. Including these

effects would increase the effect of CO by 50% for only primary, and 140% for primary and secondary aerosol effects (Shindell et al., 2009). Due to the short atmospheric lifetimes of CO (2-3 months) and $CH_4$ (12.4 years), using the short term $GWP_{20}$ would lead to a ~3 times higher impact for CO and ~2.5 times higher impact for $CH_4$ (Myhre et al., 2013). Since we assume sequestration of atmospheric $CO_2$ upon regrowth, the $GWP_{100}$ we used for CO (2.2) and $CH_4$ (32) do not include the GWP of

$CO_2$ from methane oxidation which would add roughly $(1 \times \frac{MW_{CO2}}{MW_i})$ to the $GWP_i$, depending on the considered timeframe.

In the savanna biome, fires typically occur frequently with fire return times strongly dependent on the amount of rainfall, hence productivity (Bistinas et al., 2014; Govender et al., 2006; Staver et al., 2011b). As the vegetation recovers after a fire, atmospheric $CO_2$ is captured during photosynthesis, thus balancing $CO_2$ emissions during the fire. This net-zero emission for

$CO_2$ is true for stable savanna systems with rapid regrowth while forest $CO_2$ emissions from fires take longer to be compensated for. For peat underlying humid grasslands, however, some of these emissions might be attributed to carbon that was stored over thousands of years. These carbon stocks will not regenerate at a rate that is relevant to current climate change. As peat layers are still moist in the EDS, the ratio of aboveground fuel with a short carbon cycle to long carbon-cycle peat may be seasonally dependent. Also, in the case of deforestation, $CO_2$ uptake does not balance out the loss in biosphere carbon stocks

due to the fire. Based on our measurements, we cannot conclude whether peat from the soil underlying the humid grasslands contributed to the fuel mixture. If we would not assume $CO_2$ uptake, $CO_2$-eq EFs would be 453% and 297% higher for gallery forest and peat, respectively. Our assumptions to calculate the climate impact of these fires may therefore be seen as conservative, and are only valid for stable systems.

### 4.4. N$_2$O EF uncertainty

$N_2O$ EFs were significantly lower than the 0.20 g kg$^{-1}$ that is currently used in GFED based on Akagi et al. (2011) and the 0.21 g kg$^{-1}$ for savanna in Andreae and Merlet (2001). However, the values we find are more in line with other savanna measurements from South America (0.05-0.07 g kg$^{-1}$; Hao et al., 1991; Susott et al., 1996), Australia (0.07 – 0.12 g kg$^{-1}$; Hurst et al., 1994; Meyer et al., 2012; Surawski et al., 2015) and Africa (0.16 g kg$^{-1}$; Cofer et al., 1996). The high average $N_2O$ EF in the Akagi et al. (2011) and Andreae and Merlet (2001) databases may partially be linked to the use of stainless steel sample

containers in older studies, leading to $N_2O$ formation in the sample container (Muzio and Kramlich, 1988). Due to the low concentrations and small departure from background conditions, $N_2O$ is notoriously difficult to measure. Fig. 10 shows that many EFs were negative. This occurs when concentrations in the smoke samples were below background concentrations. Although $N_2O$ is destroyed in flaming combustion (Winter et al., 1999a, 1999b) and negative emissions are thus theoretically possible, we expect it is more likely to be a measurement error. We found extremely high and low EFs in samples with low

overall EMRs. The normal Gaussian distribution pattern in Fig. 11 indicates high measurement uncertainty at low smoke concentrations. The positive and negative tails of this Gaussian error partially balance out and their weight is low relative to higher concentration measurements. Therefore, the effects of this error on the weighted average EFs should be limited. Still, a

degree of caution is advised while dealing with $N_2O$ EFs. The relative error in the 2017 campaign was higher than in the 2018 campaigns due to improvements in the algorithms used to stabilize the CO and $N_2O$ sensor. When comparing the same dataset based on vegetation type, a clear shift of the average $N_2O$ EF can be found (Fig. 11b). For vegetation types with a low number of measurements or cumulative smoke signal, the large spread reflects much higher uncertainty.

## 4.5 Limitations of the study

The findings of this study have to be seen in light of some limitations. Field measurements take place in an uncontrolled environment. This means wind conditions vary, possibly affecting the temperature and combustion efficiency of the fire and the type of fuel it consumes. Many processes happen at once during a fire making it challenging to obtain a representative EF for all stages of the process. Future research will focus on further improving the UAV-based measurement methodology to avoid possible biases as discussed in Sect. 4.2.1. We used generalized vegetation classes based on remote sensing, albeit that we lacked fuel measurements to substantiate or nuance this classification. Although the number of samples taken is substantially higher than earlier campaings, the sample size for individual categories of 'vegetation class' and 'years without fire' is in some cases small, meaning we could not always disentangle all different combinations of classes. Measuring more fires, covering a larger geographical area, and adding fuel- and wind speed measurements could provide further insights for the variability we found.

## 5. Conclusions

We obtained over 800 fresh smoke samples in different Cerrado vegetation types, during three fieldwork campaigns at various stages of the fire season. EFs of $CO_2$, CO, $CH_4$, and $N_2O$ were calculated from the difference between sample bag and background concentrations based on the carbon mass balance method. Weighted average EFs over the combined Cerrado vegetation in the EESGT study region for CO, $CH_4$ and $N_2O$ where 48 g $kg^{-1}$. 0.78 g $kg^{-1}$ and 0.11 g $kg^{-1}$ , respectively in the early dry season. In the late dry season, weighted average EFs were 41 g $kg^{-1}$ for CO (-15% compared to early dry season), 0.68 g $kg^{-1}$ for $CH_4$ (-13%) and 0.12 g $kg^{-1}$ for $N_2O$ (+17%). Apart fron the intraseasonal $N_2O$ EF decrease in grasslands and increase in typical cerrado, we did not find major intraseason EF differences that were statistically significant (p < 0.1). Some variability was explained by vegetation type, and fire history in open grasslands, whereas relative humidity only had a minor impact on variability. While we found some evidence pointing towards more efficient combustion in the LDS, the difference in weighted average EFs over the campaigns was low, while the variability during each campaign was substantial. Our findings thus imply that the effectiveness of carbon mitigation in fire abatement programs is not significantly impacted by seasonal changes in EFs for the fieldwork site and length of fire season sampled.

Overall, EFs for CO and $CH_4$ were 36% and 72% lower than EFs found in previous studies in the Cerrado and savanna fires in general. The lower $CH_4$ EFs compared to previous studies were not fully explained by higher MCE, but rather by a reduction

in the steepness of the slope of the linear regression of $CH_4$ EF as a function of MCE. We found that in our study region, $N_2O$ EFs for Cerrado vegetation were approximately half the value used in large-scale emission assessments. Uncertainties for $N_2O$ measurements are high, especially in low-concentration samples. However, these lower EFs are also found in more recent savanna studies and could indicate a substantial overestimation of the contribution of fires in the $N_2O$ budget in global databases. Seasonal effects of $N_2O$ EF were opposite for grass fuels contrasted with more shrub-dominated vegetation types. Finally, our findings indicate that accounting of CO should be considered in carbon schemes. While not a direct greenhouse gas, it has a significant effect on fire radiative forcing through its indirect effect on the $CH_4$ and $O_3$ concentration.

**Data availability**

Measurement data is available upon request.

**Author's contribution**

RV, GRvdW, and MVGA designed the study; RV and MBA conducted the experiments; RV, MMC and GRvdW participated in data analysis and/ or interpretation; RV wrote the manuscript; RV, GRvdW, ACSB edited the manuscript.

**Competing interests**

The authors declare that they have no conflict of interest.

**Acknowledgements**

This research has been supported by the Netherlands organization for Scientific Research (NWO) (Vici scheme research programme, no. 016.160.324). The Chico Mendes Institute for Conservation of Biodiversity (ICMbio) and the Center for Environmental Monitoring and Fire Management (CEMAF) led the fieldwork which would not have been possible without the rangers working at EESGT. Also, Alan Silva, Eduardo Ganassoli Neto, Micael Moreira, and Jader Nunes Cachoeira were indispensable for providing all logistics related to the fieldwork. We thank Robert Yokelson and Martin Wooster for valuable discussions on emission factor calculation. Finally, we wish to thank Anja Hoffmann for connecting the authors.

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

**Table 1: Number of samples and ancillary information about the field campaigns.**

| Vegetation class | Fractional tree cover | Avarage fire return time 2013-2018 | % of EESGT | EDS samples | LDS samples |
|---|---|---|---|---|---|
| Open grassland (campo limpo/campo sujo) | 0-5 % | 3.4 years | 17.6 % | 162 | 122 |
| Open cerrado (Cerrado ralo) | 5-20 % | 3.8 years | 35.6 % | 310 | 113 |
| Typical cerrado (Cerrado típico / cerrado sensu stricto) | 20-50 % | 4.0 years | 25.1 % | 20 | 35 |
| Gallery forest (Mata de Galeria/ Mata Ciliar) | Continuous canopy | 8.6 years | 3.0 % | 0 | 23 |
| humid grasslands (Campo limpo Úmido/Veredas) | Sparse palm trees | 3.7 years | 9.6 % | 0 | 12 |

**Table 2: Description of analysis equipment used**

| Analysis equipment | Technique | Gas species | Measurement precision | Calibration gas Concentration | Calibration gas accuracy |
|---|---|---|---|---|---|
| **Los Gatos micro-portable $CO_2$/$CH_4$ analyzer** | Off-axis integrated-cavity output spectroscopy | $CO_2$ | 2 ppmv | 4968 ppmv | 2% |
| | | $CH_4$ | 3 ppbv | 15.71 ppmv | 5% |
| **Aeris Pico mid-IR Laser-based CO/$N_2O$ analyzer** | Cavity ring-down spectroscopy | CO | 1 ppbv | 103.0 ppmv | 2% |
| | | $N_2O$ | 1 ppbv | 1.15 ppmv | 2% |

**Table 3: Weighted mean EF (g kg$^{-1}$) for various vegetation types for EDS and LDS fires. The standard error of the mean (SEM) is given in parentheses.**

| Vegetation | Season | samples | MCE | EF $CO_2$ | EF CO | EF $CH_4$ | EF $N_2O$** |
|---|---|---|---|---|---|---|---|
| Open grassland (campo limpo/campo sujo) 0-5% tree cover | EDS:<br>LDS:<br>$\Delta_{LDS-EDS}$(%) | 162<br>122 | 0.954 (0.002)<br>0.962 (0.002)<br>+0.87%* | 1662 (3)<br>1676 (5)<br>+1.0%* | 51 (1.6)<br>43 (2.0)<br>-15%* | 0.74 (0.03)<br>0.70 (0.04)<br>-6% | 0.087 (0.01)<br>0.039 (0.01)<br>-55%* |
| Open cerrado (cerrado ralo) 5-20% tree cover | EDS:<br>LDS:<br>$\Delta_{LDS-EDS}$(%) | 310<br>113 | 0.959 (0.001)<br>0.962 (0.003)<br>+0.32% | 1671 (2)<br>1677 (7)<br>+0.3% | 46 (1.2)<br>43 (3.7)<br>-6% | 0.69 (0.02)<br>0.64 (0.06)<br>-7% | 0.123 (0.01)<br>0.143 (0.02)<br>+14%* |
| Typical cerrado (cerrado típico) 20-50% tree cover | EDS:<br>LDS:<br>$\Delta_{LDS-EDS}$(%) | 20<br>35 | 0.953 (0.003)<br>0.961 (0.005)<br>+0.91% | 1657 (6)<br>1676 (10)<br>+1.2% | 52 (3.4)<br>43 (5.3)<br>-18% | 0.90 (0.04)<br>0.71 (0.10)<br>-21% | 0.109 (0.01)<br>0.147 (0.01)<br>+34%* |
| Gallery forrest (mata de galeria/mata ciliar) | EDS:<br>LDS: | 0<br>23 | -<br>0.930 (0.012) | -<br>1668 (27) | -<br>80 (13.2) | -<br>2.06 (0.43) | -<br>0.507 (0.09) |
| Humid grassland (campo limpo úmido/veredas) | EDS:<br>LDS: | 0<br>12 | -<br>0.870 (0.025) | -<br>1456 (44) | -<br>138 (25.8) | -<br>5.18 (0.17) | -<br>0.301 (0.06) |
| Cerrado $\overline{EF}$ (weighted by % BA 2013-2018) | EDS:<br>LDS:<br>$\Delta_{LDS-EDS}$(%) | | 0.957<br>0.963<br>+0.68% | 1664<br>1679<br>+1% | 48<br>41<br>-15% | 0.78<br>0.68<br>-13% | 0.105<br>0.123<br>+17% |

* Using a two-tailed t-tests with unequal variance, the difference is statistically significant (p < 0.1)

** $N_2O$ weighted average EFs and SEMs are given for samples with (>15 moles) additional carbon.

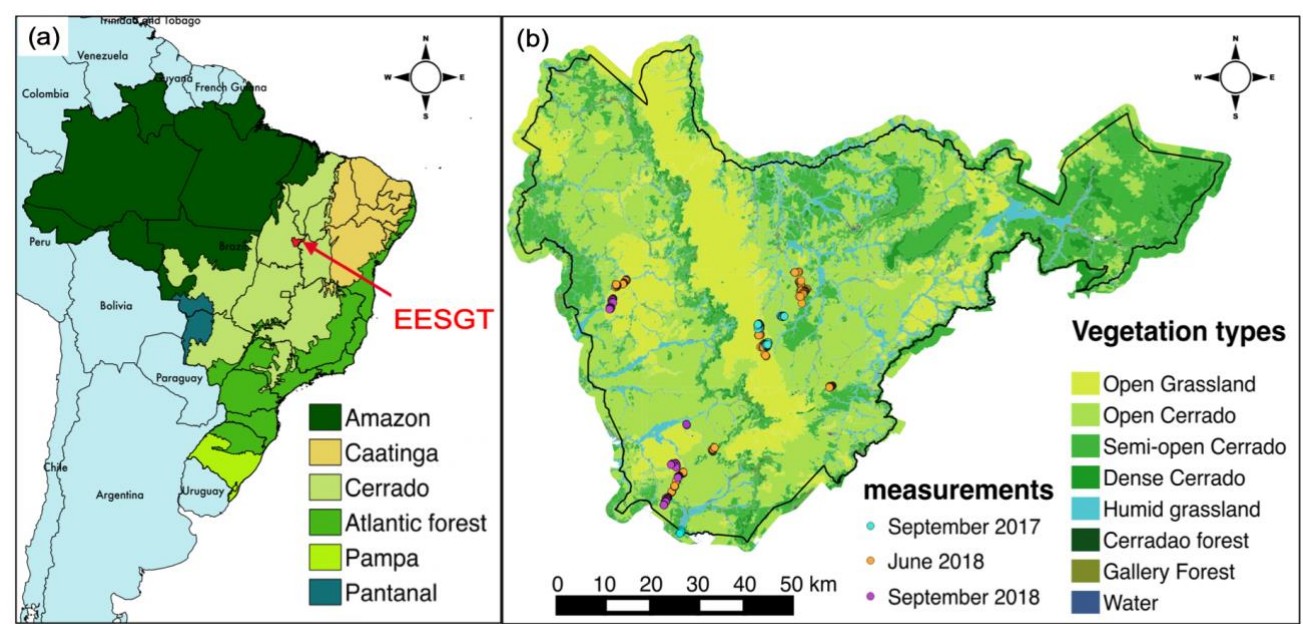

**Figure 1: a) Location of the Estação ecológica Serra Geral do Tocantins in the Cerrado biome in the Brazilian state of Tocantins. b) Vegetation types in the Estação Ecológica Serra Geral do Tocantins (Franke et al., 2018; Orozco-Filho, 2017) with the locations of the measurements.**

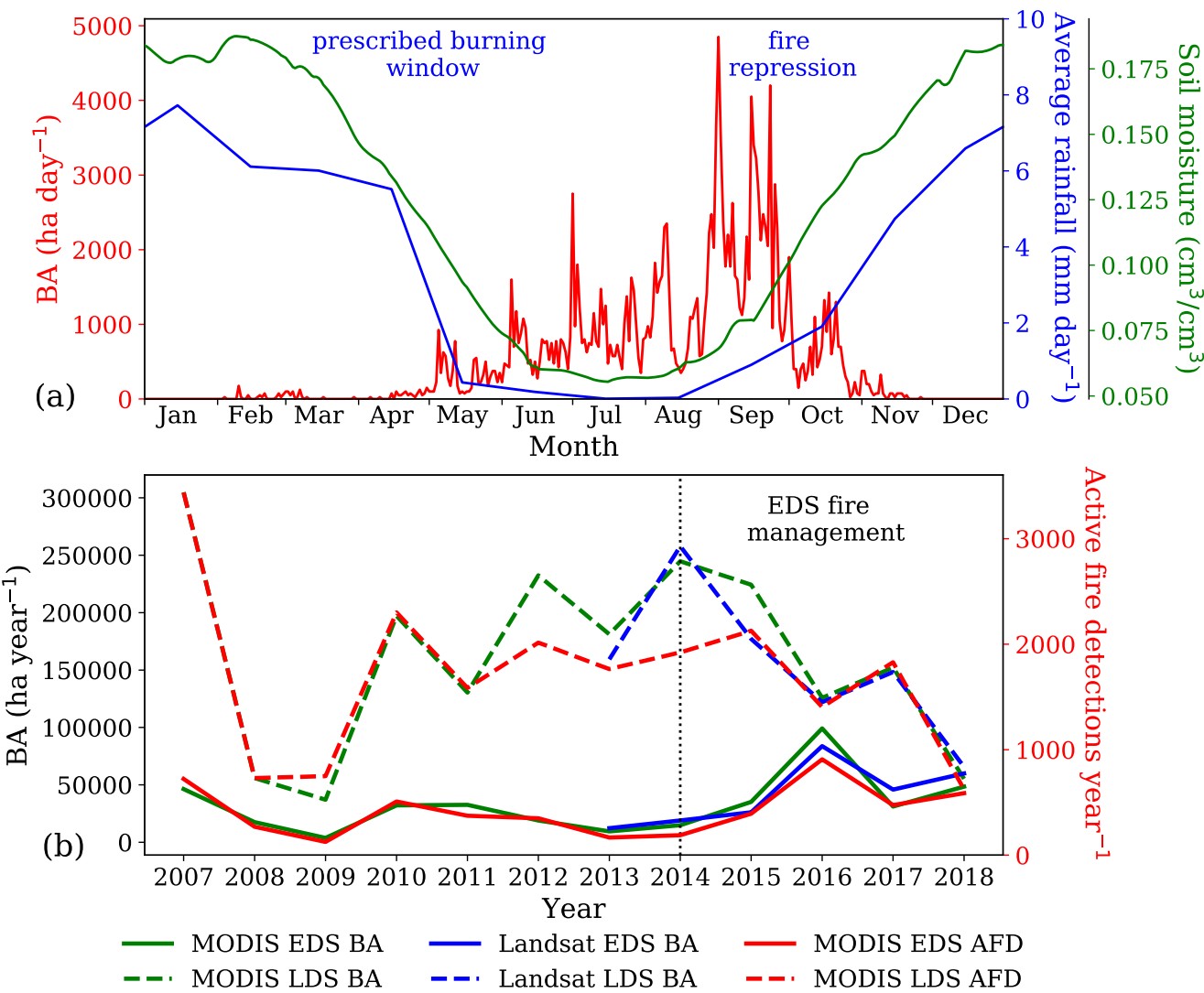

**Figure 2.** Seasonality and inter-annual variability of **(a)** Daily burned area (BA; MCD64A1 C6; Giglio et al. (2018)) as well as monthly rainfall and soil moisture, averaged over the 2013-2018 period. The prescribed burning season and repression season are hatched. **(b)** Early dry season (EDS, before July 1ˢᵗ) and late dry season (LDS, after July 1ˢᵗ). annual burned area and active fire detections (AFD, MOD14A1v6/MYD14A1v6 (Giglio and Justice, 2015)) over the 2007-2018 period.

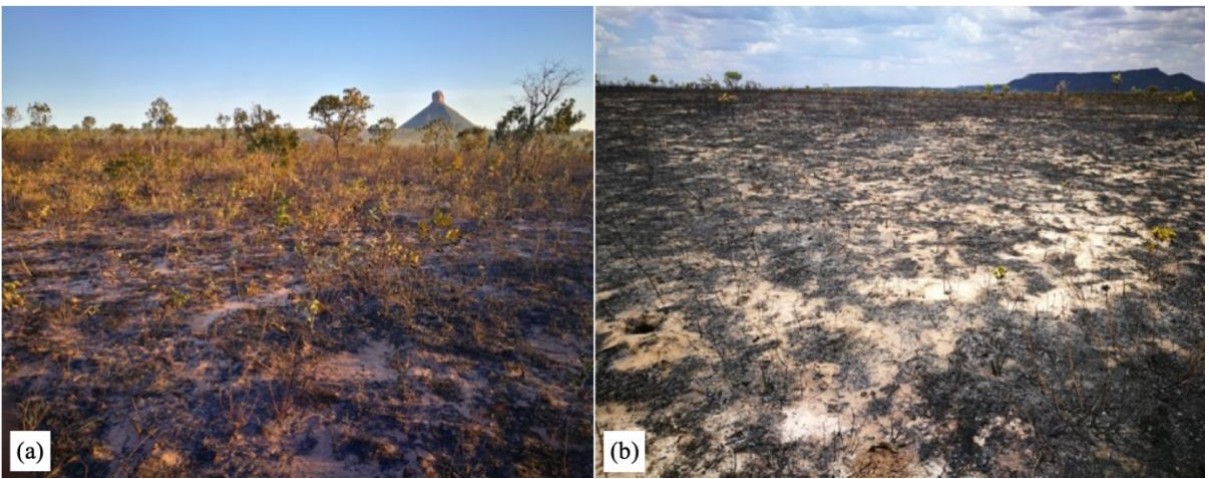

**Figure 3: Typical post-fire images showing the much smaller impact of EDS fires, in this case in June (left), compared to LDS fires in September (right).**

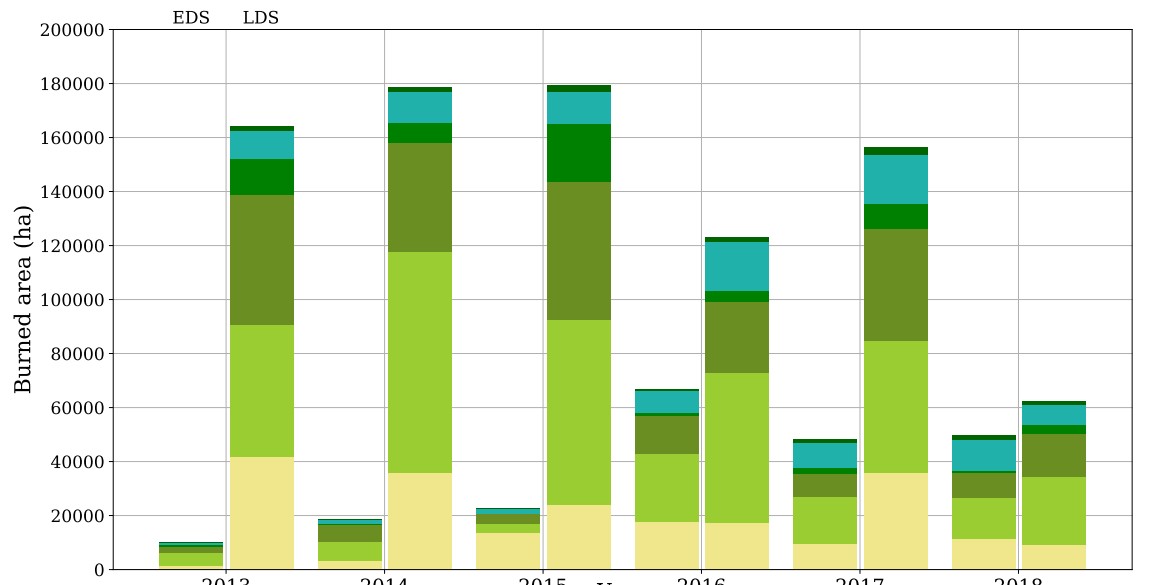

**Figure 4: Partitioning of the burned area over the EDS (before July 1st) (left columns) and LDS (after July 1st) (right columns) for the various vegetation types.**

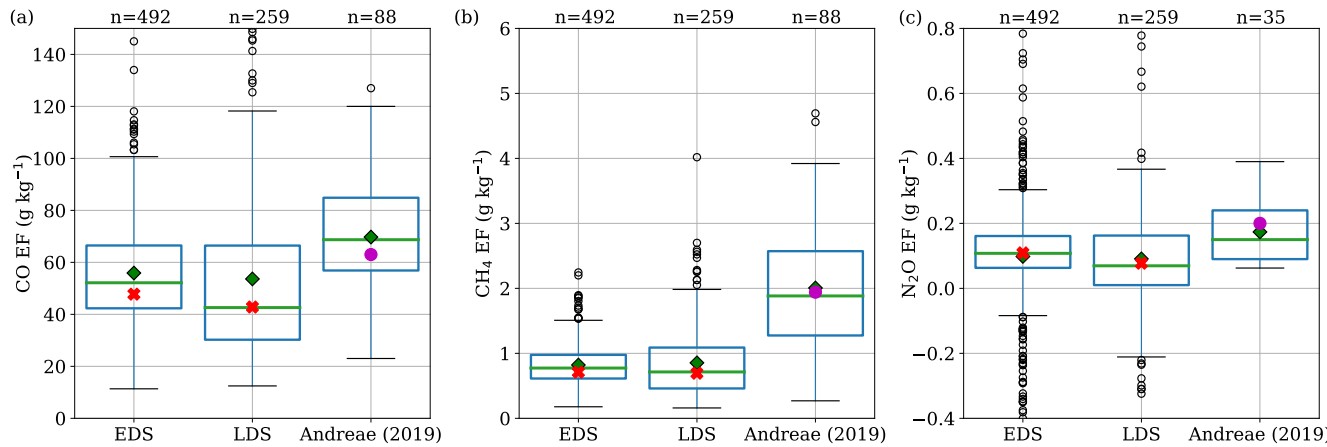

**Figure 5: EFs (g kg$^{-1}$) in the EDS and LDS as well as the EFs from savanna measurements used in the Andreae (2019) EF compilation. The green diamond represents the arithmetic mean and the red cross represents the EMR-weighted mean value. The purple dot represents the value that is used in GFED for savanna fires.**

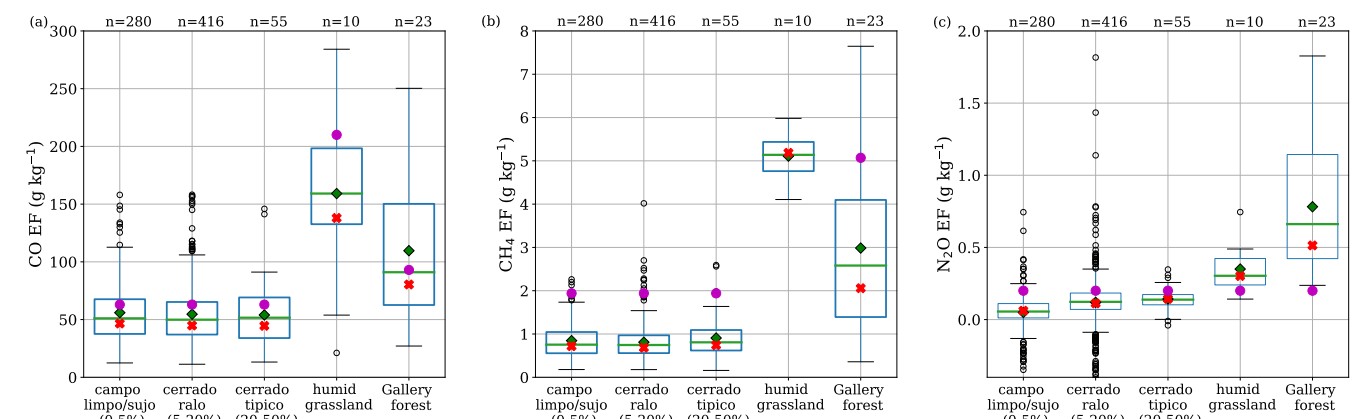

**Figure 6: EFs (g kg$^{-1}$) of CO, CH$_4$, and N$_2$O for the various vegetation types. The green diamond represents the arithmetic mean and the red cross represents the EMR-weighted mean. The purple dot represents the values that are used in GFED for 'savanna', 'peat' and 'tropical deforestation' fires respectively.**

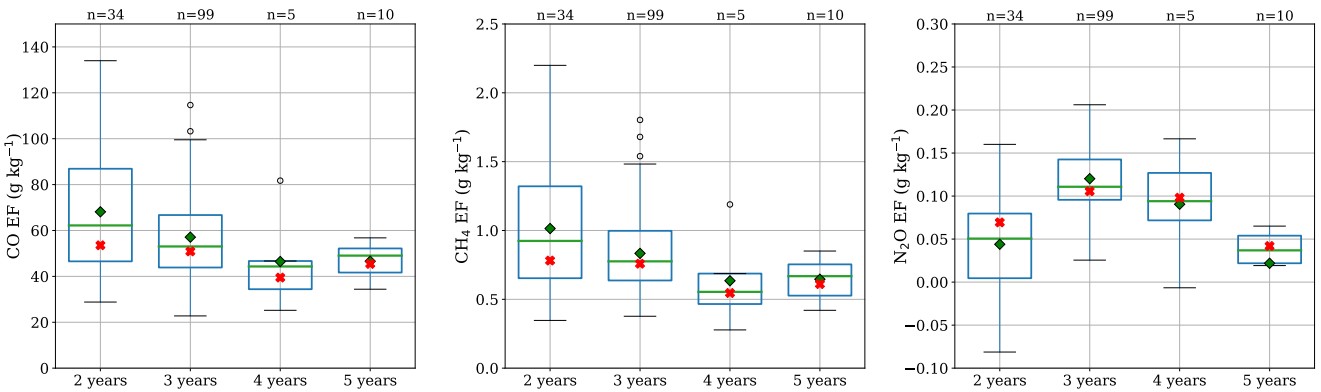

**Figure 7: EFs (g kg⁻¹) for CO, CH₄, and N₂O for open grassland samples for different periods since last fire. The green diamond represents the arithmetic mean and the red cross represents the EMR-weighted mean.**

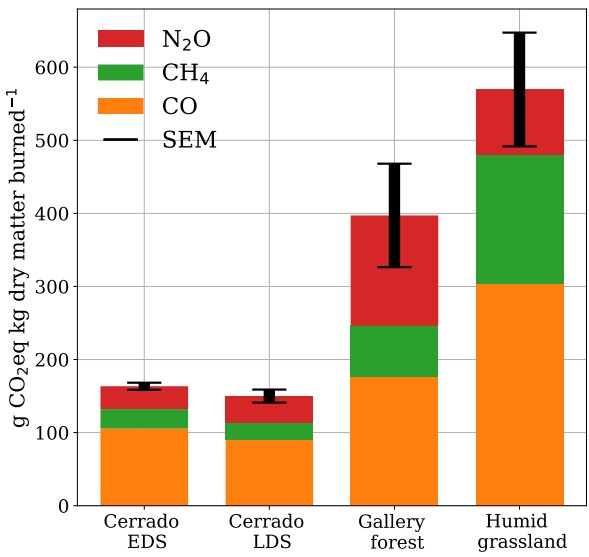

5    **Figure 8: CO₂ equivalents using GWP with a 100-year horizon and including indirect atmospheric effects for various fire types. The black errorbar represents the propagation of the standard error of the mean (SEM) of the combined CO₂eq emissions.**

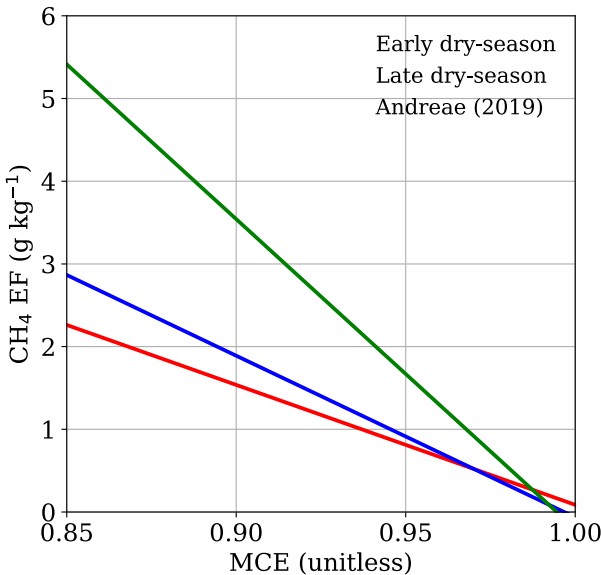

**Figure 9: Relation between the MCE and CH₄ EF for all EDS and LDS samples from Cerrado vegetation fires (i.e. excluding humid grasslands and gallery forest samples). Existing savanna measurements are shown using the study-average values in the Andreae (2019) database.**

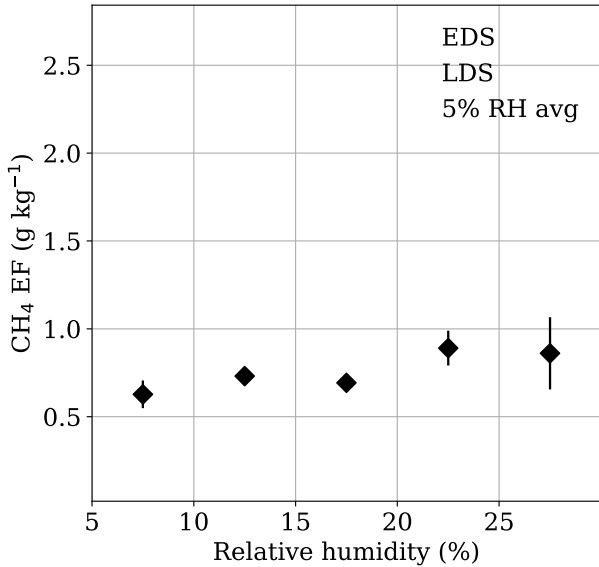

**Figure 10: CH₄ EFs as a function of relative humidity based on measurements on the UAV at the time of sampling. The size of the dots represents the ΔCH₄ EMR (ppm) in the sample and therefore depicts relative contribtion to the weighted mean. The black diamonds show the weighted average CH₄ EF for each 5% relative humidity bin. The black line represents the standard error of the class' mean.**

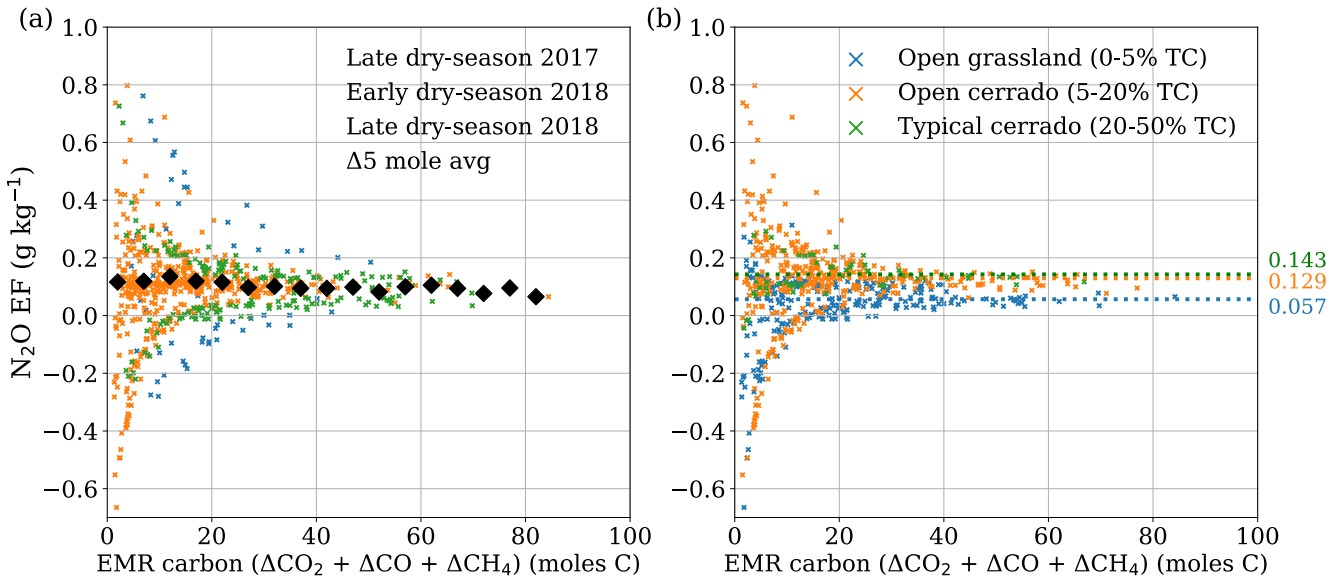

**Figure 11: N₂O EFs plotted against the cumulative EMR of the carbonaceous trace gases in the sample based on a) all cerrado measurements in the three campaigns. The black diamonds represent the averages of each 5 mole C bin. b) Combined EDS and LDS measurements in open grasslands, open cerrado and typical cerrado vegetation. The dotted lines and numbers on the right represent the weighted average N₂O EFs over all campaigns.**