# Peer review of "Intraseason variability of greenhouse gas emission factors from biomass burning in the Brazilian Cerrado"

_Biogeosciences, 2020_

## Referee Comment (RC1) · Anonymous Referee #1 · 24 May 2020

General comments: The manuscript shows results from an original dataset on biomass burning emission of green house gases at a Brazilian savanna reserve. The authors investigated the intra-seasonal variability of emission factors (EF), as well as the influence of vegetation type and fire history. Results are relevant, well presented and well discussed, and I recommend publication after a minor revision.

Specific comments:

Abstract:

* Line 17: You used the word "seasonality" along the manuscript, but I suggest the use of "intraseason variability" instead, since you are looking at the variability of emission

factors within the dry season.

* Line 21: I suggest that you include the years in which the measurements took place.

* I suggest including ranges of observed EF somewhere in the abstract.

* Lines 22-23: are these differences statistically significant? According to Table 3, they are not, considering a 95% significance level. Therefore, your conclusions should be that, overall, observations did not show a signifficant difference between EF at LDS and EDS.

Introduction:

* Page 2, Line 8: do you mean 10% of global savanna fire emissions? It is not clear in the text.

* Page 3, Lines 25-26: Are there updates on the zero-fire policy in the Brazilian cerrado? Is it still a current policy?

Methods:

* Are you aware of similar UAV-based fire emission measurements, elsewhere? If so, you may cite it, and compare the sampling strategies.

* Page 5, Line 16: here you refer to minimum daily temperatures?

* Page 5, Line 21: include a reference to Fig 1b.

* Page 5, Line 22: How was the burned area monitored? Is there a reference for the data in figure 2a?

* Page 5, Line 32: Was the RH measured at the surface? Or on board at the UAV?

* Page 6, Line 26: What was the sampling flow of the gas analyzers?

* Page 8, Line 20: Consider moving part of this paragraph to section 2.1. You might refer to Table 1 and Figure 1b (which was not referred to in the whole manuscript).

Results:

* Table 3: Include in the table caption the EF units.

* Page 9, Line 16: Where are the MCE results? I sugest that you include statistics for MCE in Table 3 or as a new boxplot in Figures 5-7.

* Page 9, Lines 21-22: What if you choose a lower significance level, for example, 90%? Would some of the differences between LDS and EDS be significant, with p<0.1?

* Page 9, Line 29 and Figure 5: Your EF values for CO and CH4 were in the lower range of previous observations at savannas (Andreae, 2019), as shown in Fig 5. Do you think that the lower EFs are characteristic of Brazilian cerrado? Or characteristic of EESGT? Please comment on that.

* Page 10, section 3.2: How about MCE? Did you observe differences related to vegetation type and fire history?

* Page 10, Line 16: Do you mean propagation of error, instead of prorogation?

* Page 10, Line 16: It would be reasonable to show the overall uncertainty on CO2-eq EF, instead of showing only N2O uncertainty, as you did in Fig 8. Also, it is not clear whether you are talking about data variability (standard deviations) or about measurement/calculation uncertainty. Please clarify.

Discussion

* Page 11, Line 2: You might state that the difference is small and not statistically significant (considering a level of significance of 95%).

* Page 11, Line 8: I miss the presentation of MCE values in your figures and tables.

* Page 11, Line 23: Fig 8 shows CO2-eq EF, and not MCE. Please check the figure reference.

* Page 11, Line 31: The lower CO and CH4 EF, as compared to the literature, is more

clearly depicted in Figure 5. I suggest that you refer to Fig 5 instead of Fig 9.

* Page 12, Line 11: What is RSC? You did not define it in the text.

* Page 12, Line 11: In this paragraph you refer to Fig.10, but I do not see a discussion about the relationship between CH4 EF and RH, which is the main feature in Fig. 10. It would be better to discuss the spread of CH4 EF during EDS and LDS based on the boxplots of Figure 5.

* Page 16, Line 8: Improvements in which software? Could this adaptation affect significantly the results and the comparison of measurements taken in 2017 and 2018?

Technical corrections:

* Check the numbering of the subitems in section 4.

* Page 15, Line 31: should refer to Fig 11 instead of Fig 10.

―――――――――――――――――

---

## Referee Comment (RC2) · Anonymous Referee #2 · 10 Nov 2020

The present manuscripts tackles an important and data-scarce topic: the seasonality of burning efficiency in open vegetation types. The authors introduce the topic by describing the biogeochemistry of fire combustion influenced by climate season, as well as set the ecological context by characterizing the fire ecology and biodiversity of the Cerrado biome, and its implication for carbon balance and related fire management. The study aims at providing new data on spatio-temporal variability of the emission factors from combusting cerrado fuel in controlled fires to update large-scale fire data bases on biomass burning. Data were collected from the smoke plumes. From the obtained data new emission factors were calculated using the state-of-the-art mass balance equation. Fuel type was derived from high-resolution vegetation type classifications using

remotely sensed data from Rapid-eye, MODIS and Landsat. Fuel amount estimated from quantifying recovery time since last fire which was derived from Landsat data. Here, the study lacks to inform the reader how this data on fuel type and fuel amount is integrated into the emission factor quantification in equ. 1 and 2, respectively. The authors need to add respective information and they need to describe how the upscaling is done in order to analyse the spatio-temporal variation.

The results describe seasonality pattern found in emission factors for N20, CO and CH4. The authors find that N20 has seasonality trends opposite to CO and CH4, where the latter indicate incomplete combustion. Statistical significance are mentioned, but not reported in detail with respective results in section 3.2. Even though it is marked in Table 3, examples should be provided in the text.

The results are then discussed in detail and contextualized using earlier publications, offering the reader to understand where earlier findings could be confirmed and where uncertainties, especially for N2O, still persist. It underlines the importance of reporting spatio-temporal variabilty in each measurement campaign also in global studies. The discussion contains a detailed description of uncertainties arising from sampling strategy, multi-day burning fires, and emission factor calculation. To avoid confusion, please also cite the original study where these numbers were taken from (it is correctly done in the methods, but worth repeating here on page 15, line 2). p. 15, lines 14-23: The discussion of the role of peat carbon contributing to carbon combustion in Cerrado fires is somewhat arbitrary, since peat combustion was not explicitly measured in these experiment, nor was the carbon storage in organic soils quantified or its proportion in the study area quantified. I would suggest to carefully discuss the wider implications of burning organic (peat) soils in the Cerrado.

The key finding of this study is clearly the fact that lower N2O emissions were found that could impact global N2O budgets if the burning conditions measured here are representative of all savannah areas which are a large contributor to global biomass burning. However, the conclusion should also contain key results (numbers) for the EF

factors for CO, CH4 and N20, incl. their uncertainty range.

Overall, the manuscript is well written, the results are substantiated and contextualized using earlier findings. The figures and table are of good graphic quality and well present the results of this study. I recommend accepting this manuscript with minor revisions.

minor corrections: p. 8, line 12: please explain BA abbreviation p.9, line 25: it should read "In Figs. 5-7 the green diamond" p. 12, line 11: explain abbreviation RSC.

---

## Author Comment (AC1) · 1 Dec 2020

We thank Reviewer #1 for the time and effort in assessing our manuscript. Please find below our point-to-point response to the review of referee #1. The revised text and updated figures are included in the updated manuscript.

1) Line 17: You used the word "seasonality" along the manuscript, but I suggest the use of "intraseason variability" instead, since you are looking at the variability of emission factors within the dry season.

While the term "seasonality" is often used in literature regarding measurements within

the dry season (Archibald et al., 2010; Hoffa et al., 1999; Meyer et al., 2012), we agree the 'intraseason variability' better captures the fact that all our measurements occur in the dry season and we have modified this throughout the manuscript, including the title.

2) Line 21: I suggest that you include the years in which the measurements took place.

We have added the years to the abstract

3) I suggest including ranges of observed EF somewhere in the abstract.

Given that we collected a very large number of samples of which a few were in non-representative humid grassland and that especially the N2O measurement has a substantial amount of noise presenting the range here would be misleading: Observed EFs ranged from 1039 to 1930 g kg-1 for CO2, 11 to 525 g kg-1 for CO, 0.1 to 7.6 g kg-1 for CH4 and -0.9 to 2.1 for N2O. Instead, we present the average and standard deviation now in the abstract to: Observed EF averages and standard deviations where 1651 (+/- 50) g kg-1 for CO2, 57.9 (+/- 28.2) g kg-1 for CO, 0.97 (+/- 0.82) g kg-1 for CH4 and 0.096 (+/- 0.174) g kg-1 for N2O.

4) Lines 22-23: are these differences statistically significant? According to Table 3, they are not, considering a 95% significance level. Therefore, your conclusions should be that, overall, observations did not show a significant difference between EF at LDS and EDS.

We agree that the statistical significance should be mentioned in the abstract and have included it in the revised abstract.

5) Page 2, Line 8: do you mean 10% of global savanna fire emissions? It is not clear in the text.

Indeed, we have revised the text to "10% of global savanna fire emissions."

6) Page 3, Lines 25-26: Are there updates on the zero-fire policy in the Brazilian cerrado? Is it still a current policy?

We changed the sentence to "until the first integrated fire management approach for some protected areas was launched in 2014, a 'zero-fire' policy had been maintained in the Brazilian Cerrado for decades"

7) Are you aware of similar UAV-based fire emission measurements, elsewhere? If so, you may cite it, and compare the sampling strategies.

To our knowledge, this is the first published study using UAV's to estimate fire emission factors. Page 5,

8) Line 16: here you refer to minimum daily temperatures?

When revising the text to address the reviewer's comment (including this one) we realised that the role of temperature is minimal and may only lead to confusion. Hence, we have excluded that sentence now

9) Page 5, Line 21: include a reference to Fig 1b.

Added a reference

10) Page 5, Line 22: How was the burned area monitored? Is there a reference for the data in figure 2a?

The burned area is calculated from MCD64A1-C6 (Giglio et al., 2018). It represents the average BA over the 2013-2018 period area within EESGT. We have added the reference to the caption.

11) Page 5, Line 32: Was the RH measured at the surface? Or on board at the UAV?

This RH is the value measured by the UAV (15m) during the background sampling. We have clarified this in the revised text.

12) Page 6, Line 26: What was the sampling flow of the gas analyzers?

For the $CO_2$ and $CH_4$ this is 1.3 L min-1, for the CO and $N_2O$ this is 0.25 L min-1. We

have added this information in the revised text in section 2.4.

13) Page 8, Line 20: Consider moving part of this paragraph to section 2.1. You might refer to Table 1 and Figure 1b (which was not referred to in the whole manuscript).

We have added references to Table 1 as well as Fig 1b. Though we agree that this also fits well with the study area description, it is also important to mention it here. We have also added a reference to Fig 1b in section 2.1.

14) Table 3: Include in the table caption the EF units.

added

15) Page 9, Line 16: Where are the MCE results? I suggest that you include statistics for MCE in Table 3 or as a new boxplot in Figures 5-7.

Since MCE is very closely related to the CO EF, we chose to only present 1 boxplot figure to avoid 2 graphs with the same information. The CO EF was chosen in our case because it is a more natural introduction to Fig. 8. The graph below compares the spread in CO EF and MCE. However, we do agree that adding the MCE is important and have added a column with MCE to Table 3 as suggested by the reviewer. Since the spread in MCE will be the same as the spread in CO EF, we don't feel that adding an additional boxplot would add much more information.

16) Page 9, Lines 21-22: What if you choose a lower significance level, for example, 90%? Would some of the differences between LDS and EDS be significant, with p<0.1?

This would not change the significance of the results. We have changed the significance level to 90% as this is more informative and changed the sentence to: "only the slight differences in open grasslands and the 14% and 34% increases in N2O EF for open cerrado and typical cerrado, respectively, were statistically significant using a two-tailed t-tests with unequal variance at a 90% significance level."

17) Page 9, Line 29 and Figure 5: Your EF values for CO and CH4 were in the lower

range of previous observations at savannas (Andreae, 2019), as shown in Fig 5. Do you think that the lower EFs are characteristic of Brazilian cerrado? Or characteristic of EESGT? Please comment on that.

Our EF's were low also compared to earlier measurements from Cerrado vegetation, particularly the CH4 EFs were low. Ferek et al. (1998) found an averaged CH4 EF of 3.7 g kg-1 and CO EF of 57 g kg-1 and Ward et al. (1992) found CH4 EFs ranging from 1-1.6 g kg-1 and CO EFs ranging from 46-70 g kg-1. This indicates that the findings may not be representable for the larger Cerrado. We have added text addressing this in section 4.2: 'Also compared to earlier measurements from Cerrado vegetation the CH4 EFs were low; Ferek et al. (1998) found an average CH4 EF of 3.7 g kg-1 and Ward et al. (1992) found CH4 EFs ranging from 1-1.6 g kg-1. This indicates that more research is needed over ideally a larger range of Cerrados and regions to understand what drives this variability.

18) Page 10, section 3.2: How about MCE? Did you observe differences related to vegetation type and fire history?

Differences in MCE would be more or less similar (though opposite) to the CO EF. In the revised manuscript we emphasized this in the text: "Fire history had some effect on the burning efficiency. We found a decrease in the CO EF and CH4 EF (and thus increase in MCE) with increasing time between fires ranging from 2 to 4 years in samples from the open grasslands (Fig. 7)." As we mentioned earlier in our response, we have also added an additional column to Table 3 with the MCE results.

19) Page 10, Line 16: Do you mean propagation of error, instead of prorogation?

Yes, corrected.

20) Page 10, Line 16: It would be reasonable to show the overall uncertainty on CO2-eq EF, instead of showing only N2O uncertainty, as you did in Fig 8. Also, it is not clear whether you are talking about data variability (standard deviations) or about measurement/calculation uncertainty. Please clarify.

We have changed Figure 8 and the error bar now represents the combined standard error of the mean (propagated into CO2-eq emissions) of all species.

We also made changes to Section 3.3: "The black error bar represents the propagation of the combined standard error of the mean for each specie to the net CO2-eq emissions. 30% to 60% of this error comes from the propagation of the uncertainty in N2O EFs."

21) Page 11, Line 2: You might state that the difference is small and not statistically significant (considering a level of significance of 95%).

We have added this to the discussion

22) Page 11, Line 8: I miss the presentation of MCE values in your figures and tables.

We have included an additional column to Table 3. As mentioned before, since the MCE would more or less be the inverted graph of the CO EF, adding 3 extra box plot graphs would not add much additional information.

23) Page 11, Line 23: Fig 8 shows CO2-eq EF, and not MCE. Please check the figure reference. You are correct, we have corrected this.

24) Page 11, Line 31: The lower CO and CH4 EF, as compared to the literature, is more clearly depicted in Figure 5. I suggest that you refer to Fig 5 instead of Fig 9.

In the revised manuscript we now refer to Fig. 5 to illustrate the lower CO and CH4 EF compared to the literature. We changed the text in the revised manuscript: 'Overall, the weighted average CO and CH4 EFs for these combined savanna fuel types were lower than most of the existing literature on savanna fires (Akagi et al., 2011; Andreae, 2019) (Fig. 5). The discrepancy with literature is particularly strong for CH4 as shown in Fig. 9 where the individual CH4 EF measurements are plotted as a function of MCE measured for the Cerrado vegetation types.'

25) Page 12, Line 11: What is RSC? You did not define it in the text. It refers to Residual Smouldering Combustion and is now spelled out in the text

26) Page 12, Line 11: In this paragraph you refer to Fig.10, but I do not see a discussion about the relationship between CH4 EF and RH, which is the main feature in Fig. 10. It would be better to discuss the spread of CH4 EF during EDS and LDS based on the boxplots of Figure 5.

We have adjusted the reference to Fig. 5 and included a reference to Fig. 10 later in the section, where we discuss the difference in CH4 EF spread compared to EMR and RH.

27) Page 16, Line 8: Improvements in which software? Could this adaptation affect significantly the results and the comparison of measurements taken in 2017 and 2018?

This improvement relates to the use of an algorithm to account for some of the background noise in the measurement. This only works when the samples are analyzed with background measurements in between long enough to identify the noise. This was not the case for the 2017 measurements. The measurement drift appears to be a random oscillation around zero, possibly related to internal heating and pressure cycles in the analyzer. When the absolute measurement is low this measurement noise may become significant (the effect will be larger in 2017 than in 2018). However, for the weighted averages this should not significantly affect the results.

28) Check the numbering of the subitems in section 4.

We resolved the section numbering problem.

29) Page 15, Line 31: should refer to Fig 11 instead of Fig 10.

Assuming you mean the Fig. 10 reference on Line 28, we changed it to Fig 11.

Please also note the supplement to this comment:
https://bg.copernicus.org/preprints/bg-2020-86/bg-2020-86-AC1-supplement.pdf

[Figure]

**Supplement:**

**Response of the authors to comments by reviewer # 1 – bg-2020-86-RC1, 2020 – "Seasonality of greenhouse gas emission factors from biomass burning in the Brazilian Cerrado"**

Roland Vernooij (corresponding author) on behalf of the authors:

We thank Reviewer #1 for the time and effort in assessing our manuscript. Please find below our point-to-point response to the review. The revised text and updated figures are included in the updated manuscript.

| Reviewer # 1 detailed comments | Author's response, reasoning and comments |
|---|---|
| Line 17: You used the word "seasonality" along the manuscript, but I suggest the use of "intraseason variability" instead, since you are looking at the variability of emission factors within the dry season. | While the term "seasonality" is often used in literature regarding measurements within the dry season (Archibald et al., 2010; Hoffa et al., 1999; Meyer et al., 2012), we agree the 'intraseason variability' better captures the fact that all our measurements occur in the dry season and we have modified this throughout the manuscript, including the title. |
| Line 21: I suggest that you include the years in which the measurements took place. | We have added the years to the abstract |
| I suggest including ranges of observed EF somewhere in the abstract. | Given that we collected a very large number of samples of which a few were in non-representative humid grassland and that especially the $N_2O$ measurement has a substantial amount of noise presenting the range here would be misleading: Observed EFs ranged from 1039 to 1930 g kg$^{-1}$ for $CO_2$, 11 to 525 g kg$^{-1}$ for CO, 0.1 to 7.6 g kg$^{-1}$ for $CH_4$ and -0.9 to 2.1 for $N_2O$. Instead, we present the average and standard deviation now in the abstract to: Observed EF averages and standard deviations where 1651 ($\pm$ 50) g kg$^{-1}$ for $CO_2$, 57.9 ($\pm$ 28.2) g kg$^{-1}$ for CO, 0.97 ($\pm$ 0.82) g kg$^{-1}$ for $CH_4$ and 0.096 ($\pm$ 0.174) g kg$^{-1}$ for $N_2O$. |
| Lines 22-23: are these differences statistically significant? According to Table 3, they are not, considering a 95% significance level. Therefore, your conclusions should be that, overall, observations did not show a significant difference between EF at LDS and EDS. | We agree that the statistical significance should be mentioned in the abstract and have included it in the revised abstract. |
| Page 2, Line 8: do you mean 10% of global savanna fire emissions? It is not clear in the text. | Indeed, we have revised the text to "10% of global savanna fire emissions." |
| Page 3, Lines 25-26: Are there updates on the zero-fire policy in the Brazilian cerrado? Is it still a current policy? | We changed the sentence to "until the first integrated fire management approach for some protected areas was launched in 2014, a 'zero-fire' policy had been maintained in the Brazilian Cerrado for decades" |
| Are you aware of similar UAV-based fire emission measurements, elsewhere? If so, you may cite it, and compare the sampling strategies. | To our knowledge, this is the first published study using UAV's to estimate fire emission factors. |

| | |
|---|---|
| Page 5, Line 16: here you refer to minimum daily temperatures? | When revising the text to address the reviewer's comment (including this one) we realized that the role of temperature is minimal and may only lead to confusion. Hence, we have excluded that sentence now |
| Page 5, Line 21: include a reference to Fig 1b. | Added a reference |
| Page 5, Line 22: How was the burned area monitored? Is there a reference for the data in figure 2a? | The burned area is calculated from MCD64A1-C6 (Giglio et al., 2018). It represents the average BA over the 2013-2018 period area within EESGT. We have added the reference to the caption. |
| Page 5, Line 32: Was the RH measured at the surface? Or on board at the UAV? | This RH is the value measured by the UAV (15m) during the background sampling. We have clarified this in the revised text. |
| Page 6, Line 26: What was the sampling flow of the gas analyzers? | For the $CO_2$ and $CH_4$ this is 1.3 L min$^{-1}$, for the CO and $N_2O$ this is 0.25 L min$^{-1}$. We have added this information in the revised text in section 2.4. |
| Page 8, Line 20: Consider moving part of this paragraph to section 2.1. You might refer to Table 1 and Figure 1b (which was not referred to in the whole manuscript). | We have added references to Table 1 as well as Fig 1b. Though we agree that this also fits well with the study area description, it is also important to mention it here. We have also added a reference to Fig 1b in section 2.1. |
| Table 3: Include in the table caption the EF units. | added |
| Page 9, Line 16: Where are the MCE results? I suggest that you include statistics for MCE in Table 3 or as a new boxplot in Figures 5-7. | Since MCE is very closely related to the CO EF, we chose to only present 1 boxplot figure to avoid 2 graphs with the same information. The CO EF was chosen in our case because it is a more natural introduction to Fig. 8. The graph below compares the spread in CO EF and MCE.

However, we do agree that adding the MCE is important and have added a column with MCE to Table 3 as suggested by the reviewer. Since the spread in MCE will be the same as the spread in CO EF, we don't feel that adding an additional boxplot would add much more information. |
| Page 9, Lines 21-22: What if you choose a lower significance level, for example, 90%? Would some of the differences between LDS and EDS be significant, with p<0.1? | This would not change the significance of the results. We have changed the significance level to 90% as this is more informative and changed the sentence to: "only the slight differences in open grasslands and the 14% and 34% increases in $N_2O$ EF for open cerrado and typical cerrado, respectively, were statistically significant using a two-tailed t-tests with unequal variance at a 90% significance level." |

| | |
|---|---|
| Page 9, Line 29 and Figure 5: Your EF values for CO and CH4 were in the lower range of previous observations at savannas (Andreae, 2019), as shown in Fig 5. Do you think that the lower EFs are characteristic of Brazilian cerrado? Or characteristic of EESGT? Please comment on that. | Our EF's were low also compared to earlier measurements from Cerrado vegetation, particularly the $CH_4$ EFs were low. Ferek et al. (1998) found an averaged $CH_4$ EF of 3.7 g $kg^{-1}$ and CO EF of 57 g $kg^{-1}$ and Ward et al. (1992) found $CH_4$ EFs ranging from 1-1.6 g $kg^{-1}$ and CO EFs ranging from 46-70 g $kg^{-1}$. This indicates that the findings may not be representable for the larger Cerrado. We have added text addressing this in section 4.2: 'Also compared to earlier measurements from Cerrado vegetation the $CH_4$ EFs were low; Ferek et al. (1998) found an average $CH_4$ EF of 3.7 g $kg^{-1}$ and Ward et al. (1992) found $CH_4$ EFs ranging from 1-1.6 g $kg^{-1}$. This indicates that more research is needed over ideally a larger range of Cerrados and regions to understand what drives this variability. ' |
| Page 10, section 3.2: How about MCE? Did you observe differences related to vegetation type and fire history? | Differences in MCE would be more or less similar (though opposite) to the CO EF. In the revised manuscript we emphasized this in the text: "Fire history had some effect on the burning efficiency. We found a decrease in the CO EF and $CH_4$ EF (and thus increase in MCE) with increasing time between fires ranging from 2 to 4 years in samples from the open grasslands (Fig. 7)." As we mentioned earlier in our response, we have also added an additional column to Table 3 with the MCE results. |
| Page 10, Line 16: Do you mean propagation of error, instead of prorogation? | Yes, corrected. |
| Page 10, Line 16: It would be reasonable to show the overall uncertainty on CO2-eq EF, instead of showing only N2O uncertainty, as you did in Fig 8. Also, it is not clear whether you are talking about data variability (standard deviations) or about measurement/ calculation uncertainty. Please clarify. | We have changed Figure 8 and the error bar now represents the combined standard error of the mean (propagated into $CO_2$-eq emissions) of all species. We also made changes to Section 3.3: "The black error bar represents the propagation of the combined standard error of the mean for each specie to the net $CO_2$-eq emissions. 30% to 60% of this error comes from the propagation of the uncertainty in $N_2O$ EFs." |
| Page 11, Line 2: You might state that the difference is small and not statistically significant (considering a level of significance of 95%). | We have added this to the discussion |
| Page 11, Line 8: I miss the presentation of MCE values in your figures and tables. | We have included an additional column to Table 3. As mentioned before, since the MCE would more or less be the inverted graph of the CO EF, adding 3 extra boxplot graphs would not add much additional information. |
| Page 11, Line 23: Fig 8 shows CO2-eq EF, and not MCE. Please check the figure reference. | You are correct, we have corrected this |
| Page 11, Line 31: The lower CO and CH4 EF, as compared to the literature, is more clearly depicted in Figure 5. I suggest that you refer to Fig 5 instead of Fig 9. | In the revised manuscript we now refer to Fig. 5 to illustrate the lower CO and $CH_4$ EF compared to the literature. We changed the text in the revised manuscript: 'Overall, the weighted average CO and $CH_4$ EFs for these combined savanna fuel types |

| | were lower than most of the existing literature on savanna fires (Akagi et al., 2011; Andreae, 2019) (Fig. 5). The discrepancy with literature is particularly strong for $CH_4$ as shown in Fig. 9 where the individual $CH_4$ EF measurements are plotted as a function of MCE measured for the Cerrado vegetation types.' |
|---|---|
| Page 12, Line 11: What is RSC? You did not define it in the text. | It refers to Residual Smouldering Combustion and is now spelled out in the text |
| Page 12, Line 11: In this paragraph you refer to Fig.10, but I do not see a discussion about the relationship between CH4 EF and RH, which is the main feature in Fig. 10.
It would be better to discuss the spread of CH4 EF during EDS and LDS based on the boxplots of Figure 5. | We have adjusted the reference to Fig. 5 and included a reference to Fig. 10 later in the section, where we discuss the difference in $CH_4$ EF spread compared to EMR and RH. |
| Page 16, Line 8: Improvements in which software? Could this adaptation affect significantly the results and the comparison of measurements taken in 2017 and 2018? | This improvement relates to the use of an algorithm to account for some of the background noise in the measurement. This only works when the samples are analyzed with background measurements in between long enough to identify the noise. This was not the case for the 2017 measurements. The measurement drift appears to be a random oscillation around zero, possibly related to internal heating and pressure cycles in the analyzer. When the absolute measurement is low this measurement noise may become significant (the effect will be larger in 2017 than in 2018). However, for the weighted averages this should not significantly affect the results. |
| Check the numbering of the subitems in section 4. | We resolved the section numbering problem. |
| Page 15, Line 31: should refer to Fig 11 instead of Fig 10. | Assuming you mean the Fig. 10 reference on Line 28, we changed it to Fig 11. |

In addition to the above-mentioned improvements we have made based on the reviewer suggestions, we have added some references to recent work that we feel improves the quality of the manuscript. Namely:

"Although no fuel moisture measurements were done during the 2018 campaigns, measurements from 2017 showed limited drying occurring from June to September, with respective average fuel moisture content declining from 63.8% to 55.4% for live grass and 11.7% to 7.2% for dead grass (Santos et al., in press)."

'The decline found in $N_2O$ EF from open grasslands that have not burned for some years (Fig. 7) may be related to the increased dead to live grass ratio of the fuel mixture as found by Santos et al. (in press).'

Akagi, S. K., Yokelson, R. J., Wiedinmyer, C., Alvarado, M. J., Reid, J. S., Karl, T., … Wennberg, P. O. (2011). Emission factors for open and domestic biomass burning for use in atmospheric models. *Atmospheric Chemistry and Physics*, *11*(9), 4039–4072. https://doi.org/10.5194/acp-11-4039-2011

Andreae, M. O. (2019). Emission of trace gases and aerosols from biomass burning – an updated assessment. *Atmospheric Chemistry and Physics*, *19*, 8523–8546. https://doi.org/10.5194/acp-19-8523-2019

Archibald, S., Scholes, R. J., Roy, D. P., Roberts, G., & Boschetti, L. (2010). Southern African fire regimes as revealed by remote sensing. *International Journal of Wildland Fire*, *19*(7), 861–878. https://doi.org/10.1071/WF10008

Ferek, R. J., Reid, J. S., Hobbs, P. V., Blake, D. R., & Liousse, C. (1998). Emission factors of hydrocarbons , halocarbons , trace gases and particles from biomass burning in Brazil. *Journal of Geophysical Research*, *103*(May 2014). https://doi.org/10.1029/98JD00692

Giglio, L., Boschetti, L., Roy, D. P., Humber, M. L., & Justice, C. O. (2018). The Collection 6 MODIS burned area mapping algorithm and product. *Remote Sensing of Environment*, *217*(March), 72–85.

https://doi.org/10.1016/j.rse.2018.08.005

Hoffa, E. A., Ward, D. E., Hao, W. M., Susott, R. A., & Wakimoto, R. H. (1999). Seasonality of carbon emissions from biomass burning in a Zambian savanna. *Journal of Geophysical Research*, *104*. https://doi.org/10.1029/1999JD900091

Meyer, C. P., Cook, G. D., Reisen, F., Smith, T. E. L., Tattaris, M., Russell-Smith, J., … Wooster, M. J. (2012). Direct measurements of the seasonality of emission factors from savanna fires in northern Australia. *Journal of Geophysical Research Atmospheres*, *117*(20). https://doi.org/10.1029/2012JD017671

Santos, M. M., Batista, A. C., Deyvid, A., Neto, E. G., Barradas, C. S., & Giongo, M. (n.d.). Characterization an dynamics of surface fuel of cerrado grassland in Jalapao region - Tocantins, Brazil. *Revista Floresta*, *51*.

Susott, R. A., Olbu, G., Baker, S. P., Ward, D. E., Kauffman, J. B., & Shea, R. W. (1996). Carbon, hydrogen, nitrogen, and thermogravimetric analysis of tropical ecosystem biomass. In J. S. Levine (Ed.), *Biomass Burning and Global Change: Remote sensing, modeling and inventory Development and Biomass Burning in Africa* (1st ed., pp. 350–360). MIT Press, Cambridge, Mass.

Ward, D. E., Susott, R. A., Kauffman, J. B., Babbitt, R. E., Cummings, D. L., Dias, B., … Setzer, A. W. (1992). Smoke and fire characteristics for cerrado and deforestation burns in Brazil : BASE-B Experiment. *Journal of Geophysical Research Atmospheres*, (December 2013). https://doi.org/10.1029/92JD01218

---

## Author Comment (AC2) · 1 Dec 2020

Response of the authors to comments by reviewer #2 – bg-2020-86-RC1, 2020 – "Seasonality of greenhouse gas emission factors from biomass burning in the Brazilian Cerrado"

Roland Vernooij (corresponding author) on behalf of the authors:

We thank Reviewer #2 for the time and effort in assessing our manuscript, and the detailed and constructive comments. Please find below our point-to-point response to the review. The revised text and updated figures are included in the updated manuscript.

1) Fuel amount estimated from quantifying recovery time since last fire which was derived from Landsat data. Here, the study lacks to inform the reader how this data on fuel type and fuel amount is integrated into the emission factor quantification in equ. 1 and 2, respectively.

In this study we do not use fuel amounts, and they are not included in Eq. (1) and Eq. (2). As they calculate the emission factor, they primarily depend on the ratio of the emitted carbonaceous species. Through the carbon content of the fuel (which does differ for different fuel types based on literature), this is then calculated back to a g kg-1 dry fuel unit. We do not attempt an estimation of the total emissions.

2) The authors need to add respective information and they need to describe how the upscaling is done in order to analyse the spatio-temporal variation.

We have added the following clarification to section 2.5:

"The weighted average emission factor (EF) for combined cerrado vegetation types in the EESGT was calculated through Eq. (3) (supplement) in which n is the number of vegetation types, BA_i is the burned area over the years 2013 to 2018 and BA_tot is the burned area over the same period. Since we lack detailed fuel load and combustion completeness data, the EF for EESGT is based on BA."

3) The results describe seasonality pattern found in emission factors for N20, CO and CH4. The authors find that N20 has seasonality trends opposite to CO and CH4, where the latter indicate incomplete combustion. Statistical significance are mentioned, but not reported in detail with respective results in section 3.2. Even though it is marked in Table 3, examples should be provided in the text.

In the revised manuscript we now refer explicitly refer to the significance of the results in the abstract, results and discussion:

in sect 3.1: "only the slight differences in open grasslands and the 14% and 34% increases in N2O EF for open cerrado and typical cerrado, respectively, were statistically

significant using a two-tailed t-tests with unequal variance at a 90% significance level."

in sect 4.1: "intraseasonal variability was smaller compared to the variability within EDS or LDS campaigns, and the difference was not statistically significant (p<0.1)

4) The results are then discussed in detail and contextualized using earlier publications, offering the reader to understand where earlier findings could be confirmed and where uncertainties, especially for N2O, still persist. It underlines the importance of reporting spatio-temporal variabilty in each measurement campaign also in global studies. The discussion contains a detailed description of uncertainties arising from sampling strategy, multi-day burning fires, and emission factor calculation. To avoid confusion, please also cite the original study where these numbers were taken from (it is correctly done in the methods, but worth repeating here on page 15, line 2).

We added the references to the discussion

5) p. 15, lines 14-23: The discussion of the role of peat carbon contributing to carbon combustion in Cerrado fires is somewhat arbitrary, since peat combustion was not explicitly measured in these experiment, nor was the carbon storage in organic soils quantified or its proportion in the study area quantified. I would suggest to carefully discuss the wider implications of burning organic (peat) soils in the Cerrado.

After closely examining the conditions under which peat burns, we decided that we cannot state with certainty that peat burned in the humid grassland fire we measured. Since the higher carbon content of 56% was based on this assumption, we have reduced this to 48% which is also used for the other cerrado species. We then recalculated the results leading to lower EFs for humid grasslands by ïA¿15%. This did not alter any of the main findings of the study. We have added the following text to the manuscript:

Sect. 4.4.2: 'The carbon content in humid grasslands is based on the assumption no peat, which has a higher carbon content of ïA¿56% (Susott et al., 1996), was combusted in the fire.'

Sect. 4.4.3: 'Based on our measurements, we cannot conclude whether peat from the soil underlying the humid grasslands contributed to the fuel mixture.'

6) The key finding of this study is clearly the fact that lower N2O emissions were found that could impact global N2O budgets if the burning conditions measured here are representative of all savannah areas which are a large contributor to global biomass burning. However, the conclusion should also contain key results (numbers) for the EF factors for CO, CH4 and N20, incl. their uncertainty range.

We added the following lines to the conclusion: 'WA EFs over the combined cerrado vegetation in EESGT for CO, CH4 and N2O where 48 g kg-1, 0.78 g kg-1 and 0.11 g kg-1 , respectively in the EDS. In the LDS, WA EFs were 41 g kg-1 for CO (-15% from EDS), 0.68 g kg-1 for CH4 (-13% from EDS) and 0.12 g kg-1 for N2O (+17% from EDS). Apart from the intraseasonal N2O EF decrease in grasslands and increase in typical cerrado, we did not find major seasonal differences that were statistically significant.'

7) p. 8, line 12: please explain BA abbreviation

It refers to burned area as observed by satellite observations. We included "burned area (BA)" in the abstract

8) p.9, line 25: it should read "In Figs. 5-7 the green diamond"

We changed it in the manuscript.

9) p. 12, line 11: explain abbreviation RSC

It refers to Residual Smouldering Combustion. This is now written out the first time we refer to the abbreviation

Please also note the supplement to this comment:
https://bg.copernicus.org/preprints/bg-2020-86/bg-2020-86-AC2-supplement.pdf

**Supplement:**

**Response of the authors to comments by reviewer #2 – bg-2020-86-RC1, 2020 – "Seasonality of greenhouse gas emission factors from biomass burning in the Brazilian Cerrado"**

Roland Vernooij (corresponding author) on behalf of the authors:

We thank Reviewer #2 for the time and effort in assessing our manuscript, and the detailed and constructive comments. Please find below our point-to-point response to the review. The revised text and updated figures are included in the updated manuscript.

| Reviewer # 2 detailed comments | Author's response, reasoning and comments |
|---|---|
| Fuel amount estimated from quantifying recovery time since last fire which was derived from Landsat data. Here, the study lacks to inform the reader how this data on fuel type and fuel amount is integrated into the emission factor quantification in equ. 1 and 2, respectively. | In this study we do not use fuel amounts, and they are not included in Eq. (1) and Eq. (2). As they calculate the emission factor, they primarily depend on the ratio of the emitted carbonaceous species. Through the carbon content of the fuel (which does differ for different fuel types based on literature), this is then calculated back to a g kg-1 dry fuel unit. We do not attempt an estimation of the total emissions. |
| The authors need to add respective information and they need to describe how the upscaling is done in order to analyse the spatio-temporal variation. | We have added the following clarification to section 2.5:

 "The weighted average ($\overline{\text{EF}}$) for combined cerrado vegetation types in the EESGT was calculated through Eq. (3) in which n is the number of vegetation types, $BA_i$ is the burned area over the years 2013 to 2018 and $BA_{tot}$ is the burned area over the same period. Since we lack detailed fuel load and combustion completeness data, the $\overline{\text{EF}}$ for EESGT is based on BA.

 $$\overline{\text{EF}} = \sum_{i=0}^{n} EF_i \times \frac{BA_i}{BA_{tot}} \quad (3)$$ " |
| The results describe seasonality pattern found in emission factors for N20, CO and CH4. The authors find that N20 has seasonality trends opposite to CO and CH4, where the latter indicate incomplete combustion. Statistical significance are mentioned, but not reported in detail with respective results in section 3.2. Even though it is marked in Table 3, examples should be provided in the text. | In the revised manuscript we now refer explicitly refer to the significance of the results in the abstract, results and discussion:

 in sect 3.1: "only the slight differences in open grasslands and the 14% and 34% increases in $N_2O$ EF for open cerrado and typical cerrado, respectively, were statistically significant using a two-tailed t-tests with unequal variance at a 90% significance level."

 in sect 4.1: "intraseasonal variability was smaller compared to the variability within EDS or LDS campaigns, and the difference was not statistically significant ($p<0.1$) |

| | |
|---|---|
| The results are then discussed in detail and contextualized using earlier publications, offering the reader to understand where earlier findings could be confirmed and where uncertainties, especially for N2O, still persist. It underlines the importance of reporting spatio-temporal variabilty in each measurement campaign also in global studies. The discussion contains a detailed description of uncertainties arising from sampling strategy, multi-day burning fires, and emission factor calculation. To avoid confusion, please also cite the original study where these numbers were taken from (it is correctly done in the methods, but worth repeating here on page 15, line 2). | We added the references to the discussion |
| p. 15, lines 14-23: The discussion of the role of peat carbon contributing to carbon combustion in Cerrado fires is somewhat arbitrary, since peat combustion was not explicitly measured in these experiment, nor was the carbon storage in organic soils quantified or its proportion in the study area quantified. I would suggest to carefully discuss the wider implications of burning organic (peat) soils in the Cerrado. | After closely examining the conditions under which peat burns, we decided that we cannot state with certainty that peat burned in the humid grassland fire we measured. Since the higher carbon content of 56% was based on this assumption, we have reduced this to 48% which is also used for the other cerrado species. We then recalculated the results leading to lower EFs for humid grasslands by ~15%. This did not alter any of the main findings of the study. We have added the following text to the manuscript:

Sect. 4.4.2:
'The carbon content in humid grasslands is based on the assumption no peat, which has a higher carbon content of ~56% (Susott et al., 1996), was combusted in the fire.'

Sect. 4.4.3:
'Based on our measurements, we cannot conclude whether peat from the soil underlying the humid grasslands contributed to the fuel mixture.' |
| The key finding of this study is clearly the fact that lower N2O emissions were found that could impact global N2O budgets if the burning conditions measured here are representative of all savannah areas which are a large contributor to global biomass burning. However, the conclusion should also contain key results (numbers) for the EF factors for CO, CH4 and N20, incl. their uncertainty range. | Added to the conclusion: 'WA EFs over the combined cerrado vegetation in EESGT for CO, CH$_4$ and N$_2$O where 48 g kg$^{-1}$, 0.78 g kg$^{-1}$ and 0.11 g kg$^{-1}$ , respectively in the EDS. In the LDS, WA EFs were 41 g kg$^{-1}$ for CO (-15% from EDS), 0.68 g kg$^{-1}$ for CH$_4$ (-13% from EDS) and 0.12 g kg$^{-1}$ for N$_2$O (+17% from EDS). Apart from the intraseasonal N$_2$O EF decrease in grasslands and increase in typical cerrado, we did not find major seasonal differences that were statistically significant.' |

| | |
|---|---|
| p. 8, line 12: please explain BA abbreviation | It refers to burned area as observed by satellite observations. We included "burned area (BA)" in the abstract |
| p.9, line 25: it should read "In Figs. 5-7 the green diamond" | We changed it in the manuscript. |
| p. 12, line 11: explain abbreviation RSC | It refers to Residual Smouldering Combustion. This is now written out the first time we refer to the abbreviation |

In addition to the above-mentioned improvements we have made based on the reviewer suggestions, we have added some references to recent work that we feel improves the quality of the manuscript. Namely:

"Although no fuel moisture measurements were done during the 2018 campaigns, measurements from 2017 showed limited drying occurring from June to September, with respective average fuel moisture content declining from 63.8% to 55.4% for live grass and 11.7% to 7.2% for dead grass (Santos et al., in press)."

'The decline found in $N_2O$ EF from open grasslands that have not burned for some years (Fig. 7) may be related to the increased dead to live grass ratio of the fuel mixture as found by Santos et al. (in press).'

---

## Author Response (AR1)

**Response of the authors to comments by reviewers – "Seasonality of greenhouse gas emission factors from biomass burning in the Brazilian Cerrado"**

Roland Vernooij (corresponding author) on behalf of the authors:

We thank both reviewers and the editor for their time and effort in assessing our manuscript, and the detailed and constructive comments which helped to improve the quality of this paper. Please find below our point-to-point response to the review. The revised text and updated figures are included in the updated manuscript. A separate 'track-changes' document is included to emphasize the changes to the manuscript.

| Reviewer # 1 detailed comments | Author's response, reasoning and comments |
|---|---|
| Line 17: You used the word "seasonality" along the manuscript, but I suggest the use of "intraseason variability" instead, since you are looking at the variability of emission factors within the dry season. | While the term "seasonality" is often used in literature regarding measurements within the dry season (Archibald et al., 2010; Hoffa et al., 1999; Meyer et al., 2012), we agree the 'intraseason variability' better captures the fact that all our measurements occur in the dry season and we have modified this throughout the manuscript, including the title. |
| Line 21: I suggest that you include the years in which the measurements took place. | We have added the years to the abstract |
| I suggest including ranges of observed EF somewhere in the abstract. | Given that we collected a very large number of samples of which a few were in non-representative humid grassland and that especially the $N_2O$ measurement has a substantial amount of noise presenting the range here would be misleading: Observed EFs ranged from 1039 to 1930 g $kg^{-1}$ for $CO_2$, 11 to 525 g $kg^{-1}$ for CO, 0.1 to 7.6 g $kg^{-1}$ for $CH_4$ and -0.9 to 2.1 for $N_2O$. Instead, we present the average and standard deviation now in the abstract to: Observed EF averages and standard deviations where 1651 ($\pm$ 50) g $kg^{-1}$ for $CO_2$, 57.9 ($\pm$ 28.2) g $kg^{-1}$ for CO, 0.97 ($\pm$ 0.82) g $kg^{-1}$ for $CH_4$ and 0.096 ($\pm$ 0.174) g $kg^{-1}$ for $N_2O$. |
| Lines 22-23: are these differences statistically significant? According to Table 3, they are not, considering a 95% significance level. Therefore, your conclusions should be that, overall, observations did not show a significant difference between EF at LDS and EDS. | We agree that the statistical significance should be mentioned in the abstract and have included it in the revised abstract. |
| Page 2, Line 8: do you mean 10% of global savanna fire emissions? It is not clear in the text. | Indeed, we have revised the text to "10% of global savanna fire emissions." |
| Page 3, Lines 25-26: Are there updates on the zero-fire policy in the Brazilian cerrado? Is it still a current policy? | We changed the sentence to "until the first integrated fire management approach for some protected areas was launched in 2014, a 'zero-fire' policy had been maintained in the Brazilian Cerrado for decades" |
| Are you aware of similar UAV-based fire emission measurements, elsewhere? If so, | To our knowledge, this is the first published study using UAV's to estimate fire emission factors. |

| | |
|---|---|
| you may cite it, and compare the sampling strategies. | |
| Page 5, Line 16: here you refer to minimum daily temperatures? | When revising the text to address the reviewer's comment (including this one) we realized that the role of temperature is minimal and may only lead to confusion. Hence, we have excluded that sentence now |
| Page 5, Line 21: include a reference to Fig 1b. | Added a reference |
| Page 5, Line 22: How was the burned area monitored? Is there a reference for the data in figure 2a? | The burned area is calculated from MCD64A1-C6 (Giglio et al., 2018). It represents the average BA over the 2013-2018 period area within EESGT. We have added the reference to the caption. |
| Page 5, Line 32: Was the RH measured at the surface? Or on board at the UAV? | This RH is the value measured by the UAV (15m) during the background sampling. We have clarified this in the revised text. |
| Page 6, Line 26: What was the sampling flow of the gas analyzers? | For the $CO_2$ and $CH_4$ this is 1.3 L $min^{-1}$, for the CO and $N_2O$ this is 0.25 L $min^{-1}$. We have added this information in the revised text in section 2.4. |
| Page 8, Line 20: Consider moving part of this paragraph to section 2.1. You might refer to Table 1 and Figure 1b (which was not referred to in the whole manuscript). | We have added references to Table 1 as well as Fig 1b. Though we agree that this also fits well with the study area description, it is also important to mention it here. We have also added a reference to Fig 1b in section 2.1. |
| Table 3: Include in the table caption the EF units. | added |
| Page 9, Line 16: Where are the MCE results? I suggest that you include statistics for MCE in Table 3 or as a new boxplot in Figures 5-7. | Since MCE is very closely related to the CO EF, we chose to only present 1 boxplot figure to avoid 2 graphs with the same information. The CO EF was chosen in our case because it is a more natural introduction to Fig. 8. The graph below compares the spread in CO EF and MCE.

However, we do agree that adding the MCE is important and have added a column with MCE to Table 3 as suggested by the reviewer. Since the spread in MCE will be the same as the spread in CO EF, we don't feel that adding an additional boxplot would add much more information. |
| Page 9, Lines 21-22: What if you choose a lower significance level, for example, 90%? Would some of the differences between LDS and EDS be significant, with p<0.1? | This would not change the significance of the results. We have changed the significance level to 90% as this is more informative and changed the sentence to: "only the slight differences in open grasslands and the 14% and 34% increases in $N_2O$ EF for open cerrado and typical cerrado, respectively, were statistically significant using a |

| | two-tailed t-tests with unequal variance at a 90% significance level." |
|---|---|
| Page 9, Line 29 and Figure 5: Your EF values for CO and CH4 were in the lower range of previous observations at savannas (Andreae, 2019), as shown in Fig 5. Do you think that the lower EFs are characteristic of Brazilian cerrado? Or characteristic of EESGT? Please comment on that. | Our EF's were low also compared to earlier measurements from Cerrado vegetation, particularly the $CH_4$ EFs were low. Ferek et al. (1998) found an averaged $CH_4$ EF of 3.7 g $kg^{-1}$ and CO EF of 57 g $kg^{-1}$ and Ward et al. (1992) found $CH_4$ EFs ranging from 1-1.6 g $kg^{-1}$ and CO EFs ranging from 46-70 g $kg^{-1}$. This indicates that the findings may not be representable for the larger Cerrado. We have added text addressing this in section 4.2: 'Also compared to earlier measurements from Cerrado vegetation the $CH_4$ EFs were low; Ferek et al. (1998) found an average $CH_4$ EF of 3.7 g $kg^{-1}$ and Ward et al. (1992) found $CH_4$ EFs ranging from 1-1.6 g $kg^{-1}$. This indicates that more research is needed over ideally a larger range of Cerrados and regions to understand what drives this variability.' |
| Page 10, section 3.2: How about MCE? Did you observe differences related to vegetation type and fire history? | Differences in MCE would be more or less similar (though opposite) to the CO EF. In the revised manuscript we emphasized this in the text: "Fire history had some effect on the burning efficiency. We found a decrease in the CO EF and $CH_4$ EF (and thus increase in MCE) with increasing time between fires ranging from 2 to 4 years in samples from the open grasslands (Fig. 7)."

As we mentioned earlier in our response, we have also added an additional column to Table 3 with the MCE results. |
| Page 10, Line 16: Do you mean propagation of error, instead of prorogation? | Yes, corrected. |
| Page 10, Line 16: It would be reasonable to show the overall uncertainty on $CO_2$-eq EF, instead of showing only $N_2O$ uncertainty, as you did in Fig 8. Also, it is not clear whether you are talking about data variability (standard deviations) or about measurement/ calculation uncertainty. Please clarify. | We have changed Figure 8 and the error bar now represents the combined standard error of the mean (propagated into $CO_2$-eq emissions) of all species.

We also made changes to Section 3.3: "The black error bar represents the propagation of the combined standard error of the mean for each specie to the net $CO_2$-eq emissions. 30% to 60% of this error comes from the propagation of the uncertainty in $N_2O$ EFs." |
| Page 11, Line 2: You might state that the difference is small and not statistically significant (considering a level of significance of 95%). | We have added this to the discussion |
| Page 11, Line 8: I miss the presentation of MCE values in your figures and tables. | We have included an additional column to Table 3. As mentioned before, since the MCE would more or less be the inverted graph of the CO EF, adding 3 extra boxplot graphs would not add much additional information. |
| Page 11, Line 23: Fig 8 shows $CO_2$-eq EF, and not MCE. Please check the figure reference. | You are correct, we have corrected this |

| | |
|---|---|
| Page 11, Line 31: The lower CO and CH4 EF, as compared to the literature, is more clearly depicted in Figure 5. I suggest that you refer to Fig 5 instead of Fig 9. | In the revised manuscript we now refer to Fig. 5 to illustrate the lower CO and $CH_4$ EF compared to the literature. We changed the text in the revised manuscript: 'Overall, the weighted average CO and $CH_4$ EFs for these combined savanna fuel types were lower than most of the existing literature on savanna fires (Akagi et al., 2011; Andreae, 2019) (Fig. 5). The discrepancy with literature is particularly strong for $CH_4$ as shown in Fig. 9 where the individual $CH_4$ EF measurements are plotted as a function of MCE measured for the Cerrado vegetation types.' |
| Page 12, Line 11: What is RSC? You did not define it in the text. | It refers to Residual Smouldering Combustion and is now spelled out in the text |
| Page 12, Line 11: In this paragraph you refer to Fig.10, but I do not see a discussion about the relationship between $CH_4$ EF and RH, which is the main feature in Fig. 10.
It would be better to discuss the spread of $CH_4$ EF during EDS and LDS based on the boxplots of Figure 5. | We have adjusted the reference to Fig. 5 and included a reference to Fig. 10 later in the section, where we discuss the difference in $CH_4$ EF spread compared to EMR and RH. |
| Page 16, Line 8: Improvements in which software? Could this adaptation affect significantly the results and the comparison of measurements taken in 2017 and 2018? | This improvement relates to the use of an algorithm to account for some of the background noise in the measurement. This only works when the samples are analyzed with background measurements in between long enough to identify the noise. This was not the case for the 2017 measurements. The measurement drift appears to be a random oscillation around zero, possibly related to internal heating and pressure cycles in the analyzer. When the absolute measurement is low this measurement noise may become significant (the effect will be larger in 2017 than in 2018). However, for the weighted averages this should not significantly affect the results. |
| Check the numbering of the subitems in section 4. | We resolved the section numbering problem. |
| Page 15, Line 31: should refer to Fig 11 instead of Fig 10. | Assuming you mean the Fig. 10 reference on Line 28, we changed it to Fig 11. |

| Reviewer # 2 detailed comments | Author's response, reasoning and comments |
|---|---|
| Fuel amount estimated from quantifying recovery time since last fire which was derived from Landsat data. Here, the study lacks to inform the reader how this data on fuel type and fuel amount is integrated into the emission factor quantification in equ. 1 and 2, respectively. | In this study we do not use fuel amounts, and they are not included in Eq. (1) and Eq. (2). As they calculate the emission factor, they primarily depend on the ratio of the emitted carbonaceous species. Through the carbon content of the fuel (which does differ for different fuel types based on literature), this is then calculated back to a g kg-1 dry fuel unit. We do not attempt an estimation of the total emissions. |
| The authors need to add respective information and they need to describe how the upscaling is done in order to analyse the spatio-temporal variation. | We have added the following clarification to section 2.5:

"The weighted average ($\overline{EF}$) for combined cerrado vegetation types in the EESGT was calculated through Eq. (3) in which n is the number of vegetation types, $BA_i$ is the burned area over the years 2013 to 2018 and $BA_{tot}$ is the burned area over the same period. Since we lack detailed fuel load and combustion completeness data, the $\overline{EF}$ for EESGT is based on BA.

$$\overline{EF} = \sum_{i=0}^{n} EF_i \times \frac{BA_i}{BA_{tot}} \quad (3)$$ " |
| The results describe seasonality pattern found in emission factors for $N_2O$, CO and $CH_4$. The authors find that $N_2O$ has seasonality trends opposite to CO and $CH_4$, where the latter indicate incomplete combustion. Statistical significance are mentioned, but not reported in detail with respective results in section 3.2. Even though it is marked in Table 3, examples should be provided in the text. | In the revised manuscript we now refer explicitly refer to the significance of the results in the abstract, results and discussion:

in sect 3.1: "only the slight differences in open grasslands and the 14% and 34% increases in $N_2O$ EF for open cerrado and typical cerrado, respectively, were statistically significant using a two-tailed t-tests with unequal variance at a 90% significance level."

in sect 4.1: "intraseasonal variability was smaller compared to the variability within EDS or LDS campaigns, and the difference was not statistically significant (p<0.1) |

| | |
|---|---|
| The results are then discussed in detail and contextualized using earlier publications, offering the reader to understand where earlier findings could be confirmed and where uncertainties, especially for $N_2O$, still persist. It underlines the importance of reporting spatio-temporal variabilty in each measurement campaign also in global studies. The discussion contains a detailed description of uncertainties arising from sampling strategy, multi-day burning fires, and emission factor calculation. To avoid confusion, please also cite the original study where these numbers were taken from (it is correctly done in the methods, but worth repeating here on page 15, line 2). | We added the references to the discussion |
| p. 15, lines 14-23: The discussion of the role of peat carbon contributing to carbon combustion in Cerrado fires is somewhat arbitrary, since peat combustion was not explicitly measured in these experiment, nor was the carbon storage in organic soils quantified or its proportion in the study area quantified. I would suggest to carefully discuss the wider implications of burning organic (peat) soils in the Cerrado. | After closely examining the conditions under which peat burns, we decided that we cannot state with certainty that peat burned in the humid grassland fire we measured. Since the higher carbon content of 56% was based on this assumption, we have reduced this to 48% which is also used for the other cerrado species. We then recalculated the results leading to lower EFs for humid grasslands by ~15%. This did not alter any of the main findings of the study. We have added the following text to the manuscript:

 Sect. 4.4.2:
 'The carbon content in humid grasslands is based on the assumption no peat, which has a higher carbon content of ~56% (Susott et al., 1996), was combusted in the fire.'

 Sect. 4.4.3:
 'Based on our measurements, we cannot conclude whether peat from the soil underlying the humid grasslands contributed to the fuel mixture.' |
| The key finding of this study is clearly the fact that lower $N_2O$ emissions were found that could impact global $N_2O$ budgets if the burning conditions measured here are representative of all savannah areas which are a large contributor to global biomass burning. However, the conclusion should also contain key results (numbers) for the EF factors for CO, $CH_4$ and $N_2O$, incl. their uncertainty range. | Added to the conclusion:
 'WA EFs over the combined cerrado vegetation in EESGT for CO, $CH_4$ and $N_2O$ where 48 g kg$^{-1}$, 0.78 g kg$^{-1}$ and 0.11 g kg$^{-1}$ , respectively in the EDS. In the LDS, WA EFs were 41 g kg$^{-1}$ for CO (-15% from EDS), 0.68 g kg$^{-1}$ for $CH_4$ (-13% from EDS) and 0.12 g kg$^{-1}$ for $N_2O$ (+17% from EDS). Apart from the intraseasonal $N_2O$ EF decrease in grasslands and increase in typical cerrado, we did not find major seasonal differences that were statistically significant.' |

| | |
|---|---|
| p. 8, line 12: please explain BA abbreviation | It refers to burned area as observed by satellite observations. We included "burned area (BA)" in the abstract |
| p.9, line 25: it should read "In Figs. 5-7 the green diamond" | We changed it in the manuscript. |
| p. 12, line 11: explain abbreviation RSC | It refers to Residual Smouldering Combustion. This is now written out the first time we refer to the abbreviation |

In addition to the above-mentioned improvements we have made based on the reviewer suggestions, we have added some references to recent work that we feel improves the overall quality of the manuscript. Namely:

"Although no fuel moisture measurements were done during the 2018 campaigns, measurements from 2017 showed limited drying occurring from June to September, with respective average fuel moisture content declining from 63.8% to 55.4% for live grass and 11.7% to 7.2% for dead grass (Santos et al., in press)."

'The decline found in $N_2O$ EF from open grasslands that have not burned for some years (Fig. 7) may be related to the increased dead to live grass ratio of the fuel mixture as found by Santos et al. (in press).'

Also, we have made a slight change to the EF calculation, to make sure it is up-to-date with recent insights and therefore consistent with future work. The conversion factor to estimate carbon in particulates was lowered from 0.097 to 0.07, which did not significantly alter the results or findings of the manuscript.

$N_2O$ EFs listed in Table 3 are now based on samples containing ($>15$ moles) of enhanced carbon concentrations, in line with the discussion in Sect. 4.4.and Fig. 11. This was the result of the reviewer request to have another critical look at the significances of the found results. Significance levels were improved by justifiably excluding these low signal values. Since relative measurement errors are much smaller and average EFs for carbonaceous species are not independent of the quantity of smoke in the sample (smouldering bags tend to be lower concentration), a similar approach would not be justifiable for those species.

In a personal correspondence with D. T. Shindell, we were informed that the GWP for CO we reference from the IPCC report 2013 does not include $CO_2$ from CO oxidation. Therefore, we incorrectly compensated for it. This was remedied in the revised version.

We truly hope that the revised manuscript is now clarified enough for the editor to be accepted for biogeosciences. We really appreciate your help on improving the readability and overall quality of our paper.